# Macrophage-derived IL-1β/NF-κB signaling mediates parenteral nutrition-associated cholestasis

Karim C. El Kasmi[1,2,8], Padade M. Vue[1,2,9], Aimee L. Anderson[1,2], Michael W. Devereaux[1,2], Swati Ghosh[1,2], Natarajan Balasubramaniyan[1,2], Sophie A. Fillon[1,2], Carola Dahrenmoeller[1,2,3], Ayed Allawzi[4], Crystal Woods[1,2], Sarah McKenna[5], Clyde J. Wright[5], Linda Johnson[6], Angelo D'Alessandro[4,7], Julie A. Reisz [4,7], Eva Nozik-Grayck[4], Frederick J. Suchy[1,2] & Ronald J. Sokol[1,2]

In infants intolerant of enteral feeding because of intestinal disease, parenteral nutrition may be associated with cholestasis, which can progress to end-stage liver disease. Here we show the function of hepatic macrophages and phytosterols in parenteral nutrition-associated cholestasis (PNAC) pathogenesis using a mouse model that recapitulates the human pathophysiology and combines intestinal injury with parenteral nutrition. We combine genetic, molecular, and pharmacological approaches to identify an essential function of hepatic macrophages and IL-1β in PNAC. Pharmacological antagonism of IL-1 signaling or genetic deficiency in CCR2, caspase-1 and caspase-11, or IL-1 receptor (which binds both IL-1α and IL-1β) prevents PNAC in mice. IL-1β increases hepatocyte NF-κB signaling, which interferes with farnesoid X receptor and liver X receptor bonding to respective promoters of canalicular bile and sterol transporter genes (*Abcc2*, *Abcb11*, and *Abcg5/8*), resulting in transcriptional suppression and subsequent cholestasis. Thus, hepatic macrophages, IL-1β, or NF-κB may be targets for restoring bile and sterol transport to treat PNAC.

[1] Department of Pediatrics, Section of Pediatric Gastroenterology, Hepatology and Nutrition, University of Colorado School of Medicine, Aurora, CO 80045, USA. [2] Pediatric Liver Center, Digestive Health Institute, Children's Hospital Colorado, Aurora, CO 80045, USA. [3] Laboratory for Gastroenterology and Hepatology (GAEN), IREC, Universite Catholique de Louvain, Belgium 1, Place de l'Université, 1348 Louvain-la-Neuve, Belgium. [4] Cardiovascular Pulmonary Research and Division of Pediatric Critical Care, Department of Pediatrics, University of Colorado School of Medicine, 12800 East 19th Avenue, Aurora, CO 80045, USA. [5] Section of Neonatology, Department of Pediatrics, University of Colorado School of Medicine, 12800 East 19th Avenue, Aurora, CO 80045, USA. [6] Department of Pathology, University of Colorado School of Medicine, Building RC-1, North Tower, 12800 East 19th Avenue, Aurora, CO 80045, USA. [7] Department of Biochemistry and Molecular Genetics, University of Colorado School of Medicine, 12801, E 17th Ave, Aurora, CO 80045, USA. [8] Present address: Department of Immunology and Respiratory, Boehringer-Ingelheim Pharma GmbH &Co. KG, Birkendorferstrasse 65, 88397 Biberach an der Riss, Germany. [9] Present address: Department of Pediatrics, Providence Pediatric Gastroenterology, University of Washington, Anchorage, AK 99515, USA. Correspondence and requests for materials should be addressed to K.C.E.K. (email: Karim.Elkasmi@ucdenver.edu) or to R.J.S. (email: Ronald.Sokol@childrenscolorado.org)

Parenteral nutrition-associated cholestasis (PNAC), also known as intestinal failure-associated liver disease, is an important and life-threatening complication of chronic parenteral nutrition (PN) therapy in infants, children, and adults with significantly compromised intestinal function and enteral feeding intolerance[1–5]. In adults, PN-associated cholestasis, hepatic steatosis, and hepatic fibrosis progress to cirrhosis within 3–5 years of chronic PN, whereas this progression to end-stage liver disease is markedly accelerated to 6–12 months in infants and young children affected by short bowel syndrome, necrotizing enterocolitis, or other causes of intestinal failure[5–11]. Thus, PNAC has historically been the leading indication for multi-visceral organ transplantation (liver, intestine, other abdominal organs) in children[5,7–13]. A major impediment to the development of effective preventative and treatment strategies has been a lack of understanding of PNAC pathogenesis. Clinical studies have suggested contributing roles of the absorption of bacterial products, such as lipopolysaccharides (LPS), from compromised, inflamed intestine, and hepatic retention of phytosterols (plant sterols), components of intravenous soybean lipid emulsions[14]. In fact, the only beneficial therapies for PNAC in children have been strategies to modify the composition and amount of intravenous lipid emulsions administered, all of which have in common a lower phytosterol intake[14–24]. To explore the mechanisms underlying PNAC, we have developed a PNAC mouse model in which mice are exposed to continuous PN infusion for 7–28 days subsequent to induction of intestinal injury with dextran sulfate sodium (DSS), recapitulating the human pathophysiology in infants and adults that develop progressive PNAC. In this model, we have demonstrated the development of cholestasis, liver injury[25,26], and hepatic macrophage accumulation, key features observed in the human infant with PNAC[27,28]. Importantly, neither intestinal injury nor PN alone were sufficient to induce PNAC in mice[25,26], corroborating the clinical observation that the most severe expression of PNAC occurs in infants with intestinal failure[29,30]. PNAC in mice was associated with reduced hepatic gene expression of canalicular transporters of bile salts (BSEP encoded by *Abcb11*), bilirubin (MRP2, encoded by *Abcc2*), and phytosterols (the heterodimer sterolin1 and sterolin2, encoded by *Abcg5* and *Abcg8*)[25,26]. In addition, increased portal vein LPS concentrations were associated with increased transcription of pro-inflammatory cytokines interleukin-6 (IL-6), tumor necrosis factor (TNF), and interleukin-1 (IL-1) in purified CD11b$^+$ hepatic macrophages[25]. Both PNAC and hepatic macrophage activation were attenuated in mice with genetic deficiency in TLR4 signaling or in wild-type mice receiving broad-spectrum enteral antibiotics[25]. Finally, PNAC, macrophage activation, and reduced expression of canalicular transporters were attenuated when PN was devoid of phytosterols, while inclusion of stigmasterol (a key phytosterol in soy lipid emulsions) into otherwise phytosterol-free PN restored liver injury, cholestasis, and macrophage activation, implicating phytosterols in PNAC pathogenesis[26].

To further elucidate the molecular and cellular mechanisms in the pathogenesis of PNAC, in the present study we combine genetic, molecular, and pharmacological approaches to demonstrate an essential role for macrophage-derived IL-1β as a key cytokine in PNAC, activating hepatocyte NF-κB, which in turn interferes with farnesoid X receptor (FXR) and liver X receptor (LXR) signaling to result in transcriptional suppression of bile and sterol transporters, culminating in cholestasis. These mechanistic insights suggest that hepatic macrophages, hepatic IL-1β or NF-κB signaling may be new targets for restoring bile and sterol transport to treat PNAC.

## Results

**Role of hepatic macrophages in PNAC.** We had demonstrated that treatment of adult C57BL/6 mice with 2.5% DSS in drinking water for 4 days followed by infusion with total PN containing soy lipid emulsion for 7–28 days (DSS/PN mice) induced cholestasis and hepatocyte injury. Furthermore DSS/PN mice had increased portal vein LPS concentrations and increased cytokine transcription in purified CD11b$^+$ liver macrophages (a finding consistent with activation of recruited macrophages)[31]. We also reported that stigmasterol activated transcription of pro-inflammatory cytokines in CD11b$^+$ liver macrophages in vivo in PNAC mice and in vitro in bone marrow-derived mouse macrophages[26]. Together, these findings suggested an important role for macrophages in PNAC[14]. We first undertook a time-course experiment and a genetic approach to assess macrophage accumulation in the PNAC model. Studies in inflammatory liver injury, including other models of cholestasis, indicate a role for recruited macrophages, which typically express CD11b and CCR2, the cognate receptor for MCP1 (encoded by *Ccl2*), a major macrophage chemo-attractant, in contrast to resident macrophages which express F4/80[32,33]. We examined hepatic mRNA expression of *Ccl2*, *Ccr2*, *Itgam* (encoding CD11b) and *Emr1* (encoding the pan macrophage marker F4/80) in DSS-pretreated mice infused with PN for 3, 7, and 14 days relative to controls (untreated chow mice and DSS-pretreated mice not PN infused). In DSS/PN mice, we found significantly increased expression of all four macrophage markers (Supplementary Fig. 1A) and that *Itgam* and *Emr1* had highest expression on day 3 of PN, prior to the onset of cholestasis (Supplementary Fig. 1A). We next employed immunohistochemistry and flow cytometry to analyze liver of DSS and DSS/PN mice at 3 days, the time point with highest transcriptional increase in *Itgam* and *Emr1*. Consistent with the gene expression data, there was an expansion of F4/80-CD11b double-positive macrophages in DSS and DSS/PN mice relative to chow mice (Supplementary Fig. 1B). Our previous report had shown increased cytokine transcription in purified CD11b$^+$ liver macrophages (consistent with activation of recruited macrophages) from DSS/PN mice[25]. Because *Ccr2*$^{-/-}$ mice lack recruitment of pro-inflammatory macrophages to the liver[34,35], we exposed *Ccr2*$^{-/-}$ mice to DSS/PN3d and found significantly reduced F4/80 staining in the liver of *Ccr2*$^{-/-}$ mice (Supplementary Fig. 1D). To determine if this reduction in hepatic macrophages in *Ccr2*$^{-/-}$ DSS/PN3d mice protected the mice from PNAC, we compared DSS/PN14d wild-type (WT) mice to *Ccr2*$^{-/-}$ mice and included control groups of DSS/chow mice, PN-only mice and unmanipulated mice (chow mice) (see Methods for detailed description of the PNAC mouse model). Cholestasis was defined by increased serum bilirubin and bile acid concentrations relative to untreated chow mice and hepatocyte injury by increased serum alanine-aminotransferase (ALT) and aspartate-aminotransferase (AST). As reported in the 7-day PNAC model[25,26], only the combination of intestinal injury with PN infusion (DSS/PN mice) resulted in PNAC in this 14-day model in WT mice (Fig. 1a). Concomitant with biochemical cholestasis, mRNA expression of *Abcb11* and *Abcc2* was significantly suppressed in DSS/PN14d mice (Fig. 1a). Similarly, mRNA expression of *Nr1h4*, encoding the transcription FXR that transactivates *Abcb11* and *Abcc2*[36,37], was suppressed in WT DSS/PN mice (Fig. 1b). Consistent with findings in the DSS/PN3d mice, hepatic expression of genes representing recruited and inflammatory macrophages (*Ccr2*, *Itgam* (encoding CD11b), and *Ly6c*) was significantly increased in PNAC mice (Fig. 1c) and immunohistochemistry with F4/80, a pan macrophage marker[38], revealed markedly increased numbers of enlarged (as a sign of activation) macrophages in livers from WT PNAC mice. These biochemical, histochemical, and molecular findings in DSS/

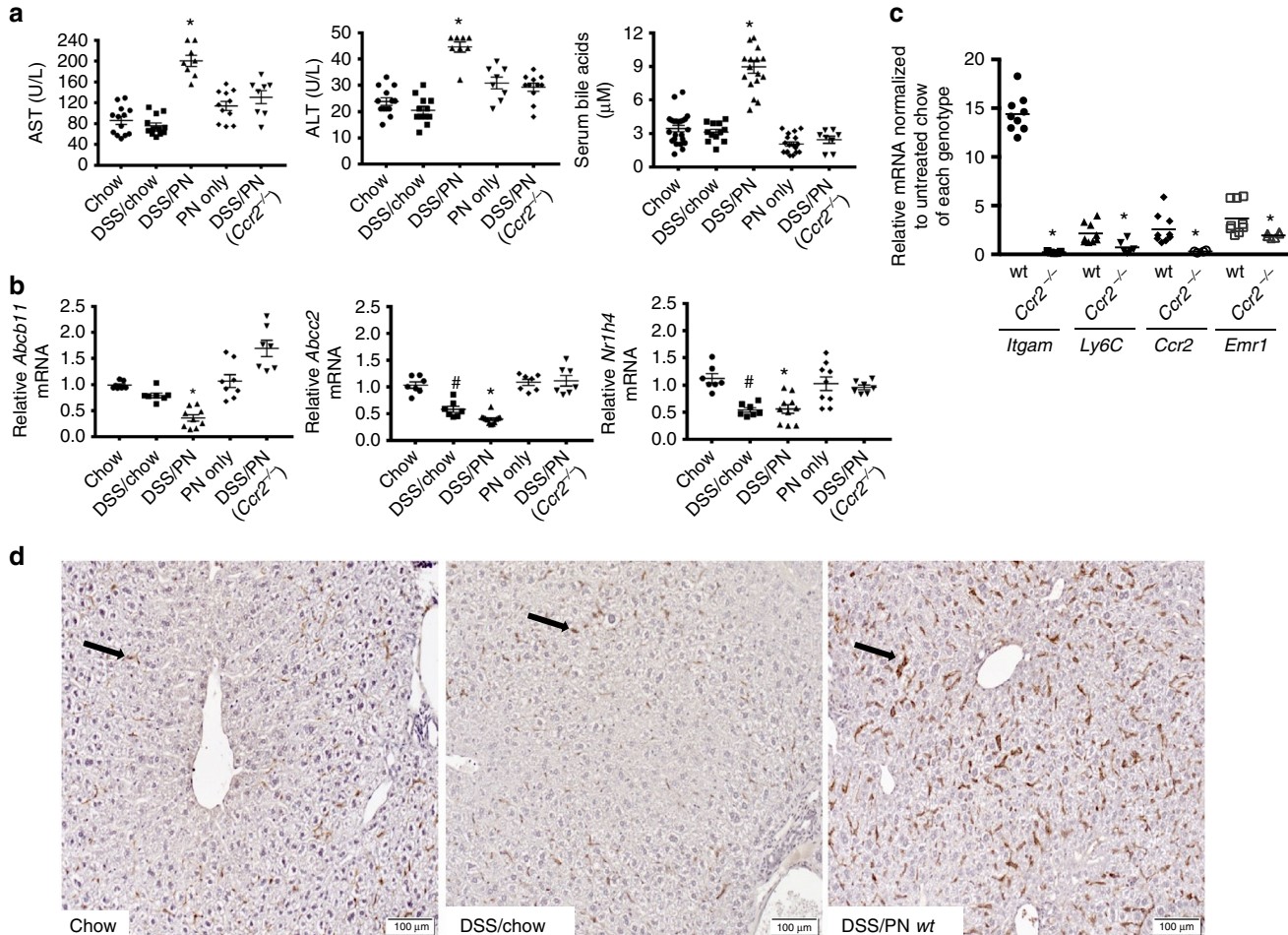

**Fig. 1** Mice with genetic *Ccr2* deficiency are protected from PNAC. **a** Peripheral serum aspartate-aminotransferase (AST), alanine-aminotransferase (ALT), and total bile acid concentrations in completely unmanipulated mice (chow), DSS-pretreated mice without any further manipulation, (DSS/chow) wild-type (WT) DSS-pretreated mice infused with PN (DSS/PN), PN infused wild-type mice without DSS pretreatment (PN only), and *Ccr2*$^{-/-}$ DSS-pretreated mice infused with PN (*Ccr2*$^{-/-}$ DSS/PN). **b** Liver mRNA from the same groups of mice as in **a** of genes encoding BSEP (*Abcb11*), MRP2 (*Abcc2*), and FXR (*Nr1h4*) and **c** liver mRNA from DSS/PN WT and *Ccr2*$^{-/-}$ mice of genes encoding CD11b (*Itgam*), Ly6C, CCR2, and F4/80 (*Emr1*). **d** Representative liver immunohistochemistry for F4/80 from chow, DSS/chow, and DSS/PN mice (black arrows point to F4/80$^+$ cells). *$p < 0.05$ vs. all other groups by one-way ANOVA and Tukey's correction (**a** and **c**). *$p < 0.05$ vs. chow, PN only, *Ccr2*$^{-/-}$ DSS/PN; #$p < 0.05$ vs. chow by one-way ANOVA and Tukey's correction (**b**). Data are depicted from individual mice for serum analysis and means from PCR triplicates from individual mice for gene expression data ±SEM

PN14d mice were markedly suppressed in *Ccr2*$^{-/-}$ DSS/PN14d mice (Fig. 1d and Supplementary Fig. 1C, D), indicating that CCR2$^+$ macrophages recruited to the liver were a critical factor for development of PNAC. Based on these observations, we hypothesized that inflammatory pathways associatsed with macrophage activation suppressed expression of bile transporters accounting for biochemical cholestasis and liver injury.

**IL-1β suppresses bile transporter expression in PNAC**. To test this hypothesis, we examined if there was a temporal relationship between development of liver injury, cholestasis, decreased expression of canalicular transporters, and increased hepatic cytokine expression. Eight-week-old male C57BL/6 mice pretreated with DSS (as in Fig. 1) were randomized into groups receiving PN for 3, 7, and 14 days. Control groups treated for 3, 7, and 14 days included DSS/chow, PN-only mice, and chow-fed mice. Serum total bilirubin and bile acids were not altered in the 3-day DSS/PN mice, but were elevated at 7 days and more pronounced at 14 days (Fig. 2a). Similarly, AST and ALT were not

elevated in 3-day DSS/PN mice but were significantly elevated after 7 and 14 days of PN (Fig. 2a). Neither cholestasis nor hepatocyte injury developed in PN-only or DSS/chow mice at any time point (Fig. 2a). Hepatic mRNA expression of *Abcb11* and *Abcc2* was suppressed in DSS/PN mice at 3 days, preceding evidence of cholestasis and hepatocyte injury, and were progressively suppressed by 14 days (Fig. 2b). Interestingly, in DSS/chow mice there was moderate suppression of *Abcb11* mRNA at the 14-day time point and suppression of *Abcc2* mRNA at the 7-day and 14-day time points (Fig. 2b), yet, importantly, this reduced transporter expression alone was insufficient to cause cholestasis.

We next measured mRNA expression of *Il1b*, *Tnfa*, and *Il6* (shown by us to be upregulated in hepatic macrophages in mice with PNAC[25,26]) by quantitative PCR (qPCR) in both livers and in isolated intrahepatic mononuclear cells (IHMCs) at these time points. Expression of *Il1b* mRNA was modestly increased in IHMCs from DSS/chow mice relative to chow controls at day 3 and declined back to control levels by day 14. In contrast, *Il1b* transcription in IHMCs from DSS/PN mice was increased significantly higher at day 3 compared to DSS/chow mice and

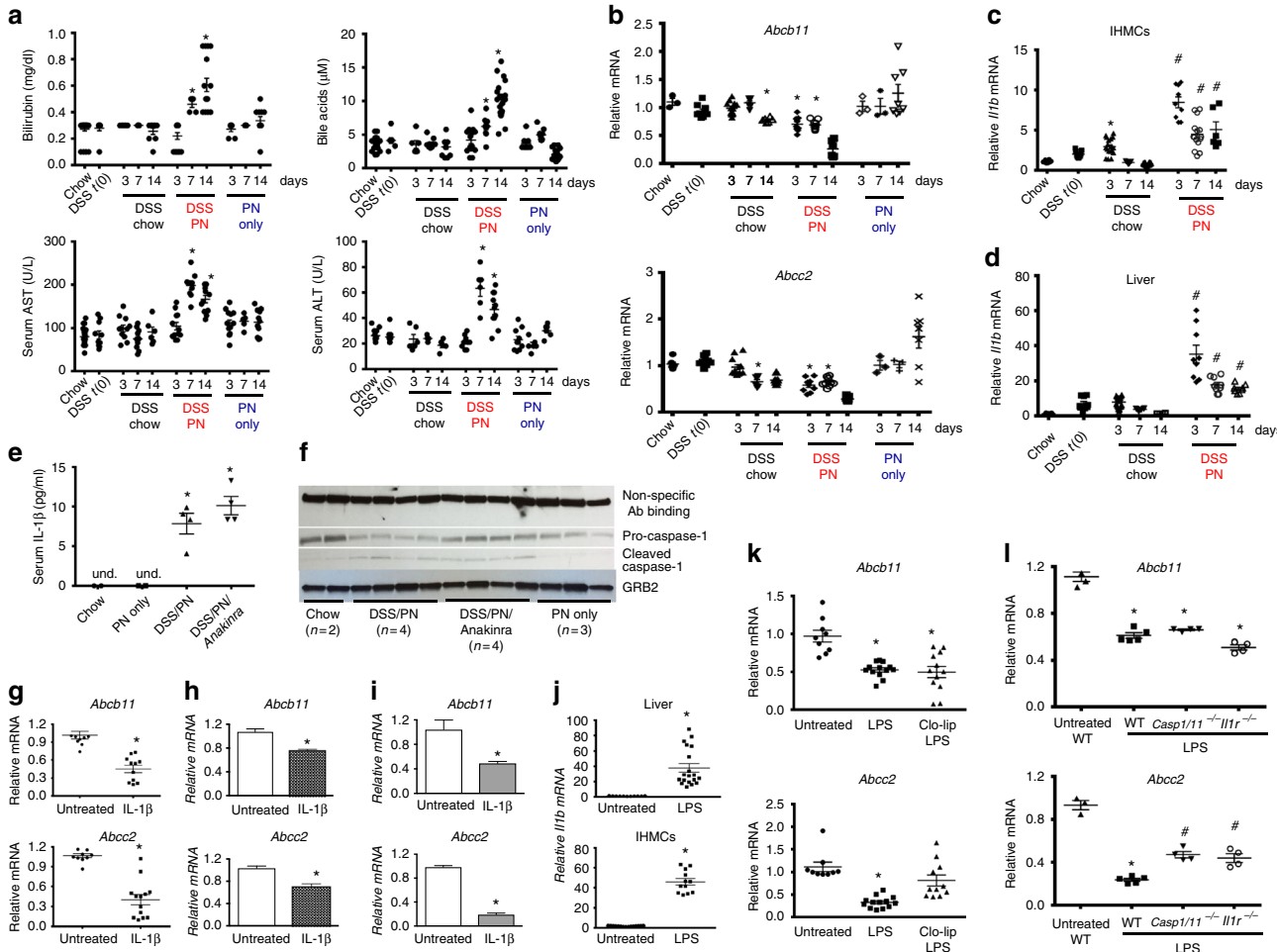

**Fig. 2** IL-1β precedes cholestasis and suppresses bile transporters. **a** Peripheral serum, AST, ALT, total bile acid, and bilirubin concentrations in chow mice, DSS mice immediately after completion of DSS pretreatment (DSS $t(0)$ mice), DSS/chow, DSS/PN, and PN only mice for 3, 7, and 14 days. **b** Liver mRNA (same mice as in **a**) of $Abcb11$ and $Abcc2$ and in purified intrahepatic mononuclear cells (IHMCs) (**c**) compared to liver homogenate (**d**) from chow, DSS $t$ (0), DSS/chow, and DSS/PN mice. **e** Peripheral serum IL-1β concentrations. **f** Immunoblot on liver homogenate for procaspase-1 and cleaved caspase-1; protein bands due to non-specific binding of this antibody and for anti-GRB2 antibody were used to show equal protein loading. For the latter, the blot needed to be stripped due to proximity in size of GRB2 (25 kDa) and IL-1β (17 kDa active, 31 kDa pro-form). **g** Liver mRNA of $Abcb11$ and $Abcc2$ from untreated and IL-1β-injected mice after 4 h in $Abcb11$ mRNA and $Abcc2$ mRNA in HuH7 cells (**h**) or primary mouse hepatocytes (**i**) left untreated or exposed to IL-1β (10 ng/ml) for 4 h. For **h** and **i** representative data are shown depicting mean ± SEM of technical triplicates from one out of at least five experiments. **j**, **k** $Il1b$, $Abcb11$, and $Abcc2$ mRNA expression in liver homogenate or in IHMCs from untreated and LPS injected mice after 4 h with or without prior (16 h) i.p. clodronate-liposome (Clo-lip) injection and in WT, $Casp1/11^{-/-}$, or $Il1r^{-/-}$ mice (**l**). *$p < 0.05$ vs. all other groups by one-way ANOVA and Tukey's correction (**a**, **b**, **e**). *$p < 0.05$ vs. chow, and #$p < 0.05$ vs. chow/DSS and chow (**c**, **d**). *$p < 0/05$ vs. untreated by $t$ test (**g**, **h**, **i**, **j**); *$p < 0.05$ vs. untreated and LPS-Clo-lip one-way ANOVA and Tukey's correction (**k**); *$p < 0.05$ vs. untreated, and #$p < 0.05$ vs. WT by one-way ANOVA and Tukey's correction (**l**). Data are depicted as individual mice for serum analysis and as means from PCR triplicates from individual mice for gene expression data (mean ± SEM)

remained elevated through day 14 (Fig. 2c). Similarly, $Il1b$ mRNA was modestly increased in liver homogenate in DSS/chow mice at day 3 and declined back to chow control levels by day 7, while elevated $Il1b$ mRNA was robust and sustained through day 14 in DSS/PN (Fig. 2d). Expression of $Il6$ was not significantly different from controls (not shown). The increased hepatic $Il1b$ mRNA in DSS/PN14d mice was accompanied by markedly elevated serum concentrations of IL-1β, which were undetectable in PN-only and DSS/chow mice (Fig. 2e). Finally, expression of hepatic cleaved caspase-1 protein, essential for production of active IL-1β[39], was increased in DSS/PN14d mice, with a concomitant reduction in procaspase-1 protein, but not in PN-only mice (Fig. 2f).

The observation that increased transcription of $Il1b$ at 3 days preceded elevated serum bilirubin, bile acids, AST and ALT suggested a causative role for IL-1β in the development of

cholestasis (Fig. 2). To determine if IL-1β directly suppressed hepatic $Abcb11$ and $Abcc2$ mRNA in vivo, we injected otherwise untreated WT mice with 200 ng/mouse recombinant IL-1β intraperitoneal (i.p.). Compared to untreated mice, within 4 h of IL-1β exposure there was significant hepatic transcriptional suppression of $Abcb11$ and $Abcc2$ (Fig. 2g). Similarly, in vitro incubation of primary mouse hepatocytes (Fig. 2h) or the human hepatocyte cell line HuH7 (Fig. 2i) with IL-1β resulted in attenuated transcription of both $ABCB11/Abcb11$ and $ABCC2/Abcc2$. These data were consistent with a role for IL-1β suppression of canalicular transporters in PNAC.

We next determined if LPS, which we had shown to be increased in portal venous blood in DSS/PN cholestatic mice[25,26], was an upstream signal for inducing macrophage $Il1b$ transcription and suppression of hepatic bile transporters. Intraperitoneal

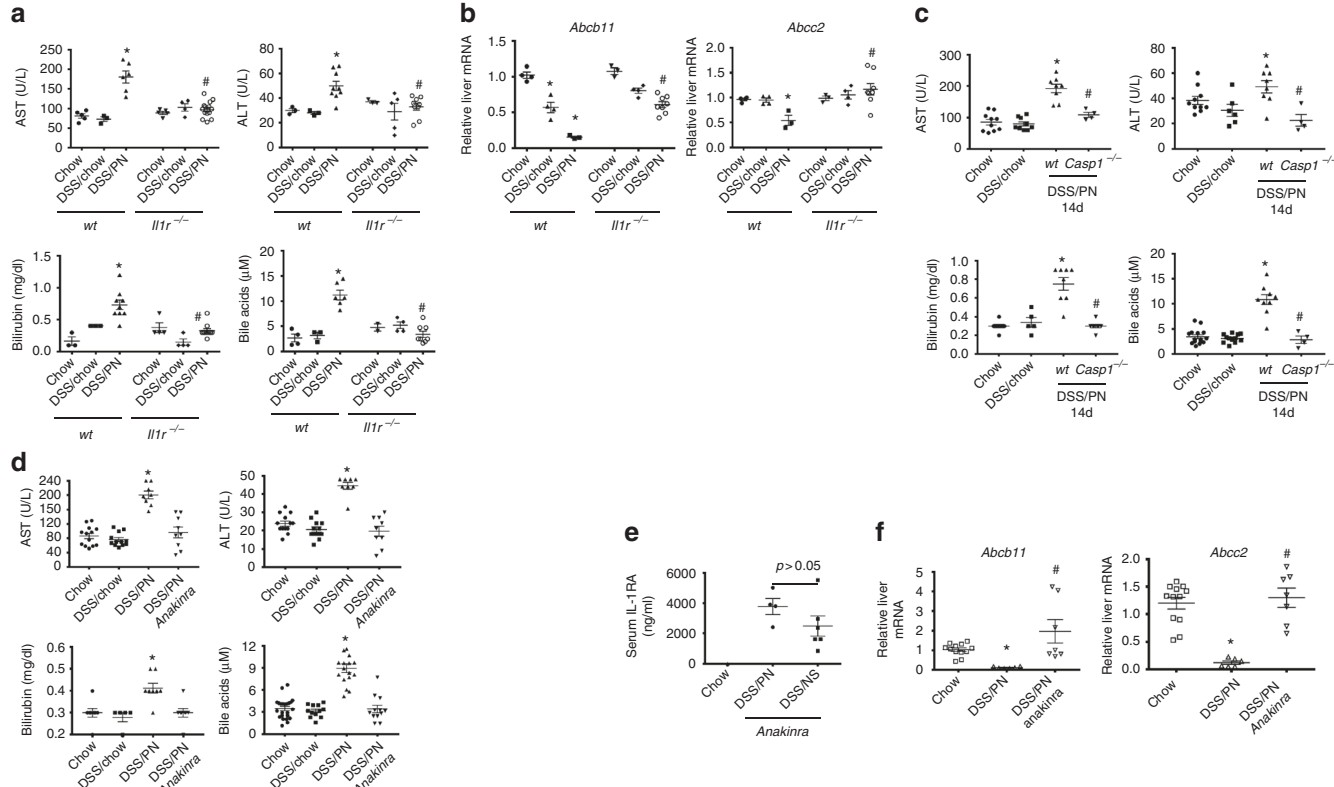

**Fig. 3** Genetic and pharmacological blockade of IL-1 prevents PNAC. **a**, **c**, **d** Peripheral serum concentrations of bilirubin, AST, ALT, and total bile acids in $Il1r^{-/-}$-treated, $Casp1^{-/-}$-treated, and Anakinra-treated DSS/PN mice relative to that in mice that were chow-fed, DSS/chow, or DSS/PN treated. *$p < 0.05$ vs. all other groups for WT mice, and #$p < 0.05$ knockout or Anakinra DSS/PN vs. DSS/PN wt; one-way ANOVA and Tukey's correction. **b**, **f** Liver mRNA for *Abcb11* and *Abcc2* in Anakinra-treated and in $Il1r^{-/-}$, DSS/PN mice relative to controls. *$p < 0.05$ DSS/PN vs. DSS/chow and chow, *$p < 0.05$ DSS/chow vs. chow; #$p < 0.05$ DSS/PN wt vs. DSS/PN $Il1r^{-/-}$; one-way ANOVA and Tukey's correction. **e** Peripheral serum concentrations of IL-1 receptor antagonist (Anakinra) in completely unmanipulated chow, Anakinra-treated DSS/PN, and Anakinra-treated DSS/NS (normal saline infused) mice. *$p < 0.05$ DSS/PN vs. chow, and #$p < 0.05$ DSS/PN vs. Anakina DSS/PN; one-way ANOVA and Tukey's correction

injection of 2.5 mg/kg LPS into WT mice resulted in increased *Il1b* transcription in both liver and isolated IHMCs (Fig. 2j) coinciding with suppression of hepatic *Abcb11* and *Abcc2* mRNA (Fig. 2k). Macrophage depletion was achieved in a separate group of mice by treatment with clodronate-liposome (Clo-lip), confirmed by liver F4/80 immunohistochemistry and reduction of mRNA for *Emr1*, the gene encoding F4/80 (Supplementary Fig. 1E, F). In macrophage-depleted mice, LPS injection failed to both increase hepatic *Il1b* transcription (Supplementary Fig. 1G) and to suppress *Abcc2* (Fig. 2k). In addition, LPS injection in mice with interrupted IL-1β signaling ($Il1r^{-/-}$ or $Casp1,11^{-/-}$ mice) was associated with partially restored *Abcc2* expression while there was no effect on *Abcb11* expression (Fig. 2l). Furthermore, in vitro treatment of HuH7 cells or primary mouse hepatocytes with LPS for 4 h did not directly suppress either *Abcb11* and *Abcc2* mRNA expression (Supplementary Fig. 1H, I). Finally, it has been suggested that IL-1β can promote TNF expression[40–42] and that blocking IL-1 receptor could reduce TNF[43] production. We therefore examined mRNA expression of *Tnfa* in liver homogenate and in isolated IHMCs 4 h after LPS injection and found that LPS was less effective in inducing *Tnfa* expression in both liver and purified IHMCs of $Il1r^{-/-}$ mice compared to WT mice, consistent with the notion that IL-1β signaling can function upstream of TNFa in a feedforward mechanism (Supplementary Fig. 1J, K). Together, these findings supported the hypothesis that in mice with PNAC, liver macrophages were activated by LPS to produce IL-1β, which, in turn, activated hepatocyte pathways that

suppressed canalicular bile acid and bilirubin transporter expression.

**IL-1β signaling is required for PNAC development.** To further determine if IL-1β played a causative role in PNAC, we employed mice with genetic deficiency in the IL-1 receptor ($Il1r^{-/-}$), which are capable of IL-1β production but unresponsive to IL-1β and IL-1α signaling[44,45]. Compared to WT DSS/PN14d mice, $Il1r^{-/-}$ DSS/PN14d mice were protected from developing PNAC (Fig. 3a) and hepatic *Abcc2* and *Abcb11* mRNA were significantly increased (Fig. 3b). Since $Il1r^{-/-}$ mice are unresponsive to both IL-1β and IL-1α[44,45], we next used $Casp1/11^{-/-}$ mice (incapable of generating mature IL-1β)[39] to isolate a role for IL-1β and observed that $Casp1/11^{-/-}$ DSS/PN14d mice did not develop PNAC (Fig. 3c). We next used pharmacological blockade of IL-1β as a complementary approach to interrogate IL-1β signaling. Treatment with intravenous anakinra (Kineret®, Amgen, Thousand Oaks, CA), a commercially available IL-1R antagonist with significant activity in both mice and humans[45], at doses of 25 mg/kg[45] on days 3 and 10 of PN completely prevented PNAC in WT DSS/PN14d mice (Fig. 3d). Anakinra serum levels at 14 days measured by enzyme-linked immunosorbent assay (ELISA)[45] were comparable to those that had been reported to be sufficient to suppress IL-1 receptor signaling[45] (Fig. 3e). Anakinra treatment of DSS/PN mice prevented the reduction in hepatic gene expression of *Abcb11* and *Abcc2* observed in DSS/PN mice treated with vehicle (Fig. 3f).

We further examined gene expression of hepatic *Nr0b2* (encoding small heterodimer partner (SHP)) and *Cyp7a*. Induction of SHP by FXR is an important negative regulator of bile acid synthesis through suppression of *Cyp7a*, which encodes the rate limiting enzyme in bile acid synthesis. In cholestasis, bile acids that would accumulate in the hepatocyte should transactivate FXR, increasing SHP which inhibits bile acid synthesis to protect the hepatocyte from the toxicity of bile acids. In WT DSS/PN mice, *Nr0b2* expression was unexpectedly markedly reduced in association with moderate reduction (40%) of *Cyp7a* mRNA relative to chow mice. In contrast, in *Il1r*$^{-/-}$-treated, Anakinra-treated and *Ccr2*$^{-/-}$-treated DSS/PN mice, *Nr0b2* expression was near normal or induced relative to WT DSS/PN mice in conjunction with stronger suppression of *Cyp7a* mRNA (80–90%) (Supplementary Fig. 2A, B). To assess the functional relevance of these gene expression data, we measured serum concentrations of taurine-conjugated bile acids by liquid chromatography-mass spectrometry (LC-MS). Indeed, serum levels of the major taurine-conjugated bile acids were significantly increased in WT DSS/PN mice relative to chow mice, while serum bile acid levels were significantly reduced in *Il1r*$^{-/-}$ and *CCr2*$^{-/-}$ DSS/PN mice, reflecting the *Cyp7a* expression data (Supplementary Fig. 2C).

Finally, we undertook experiments to verify that the observed protection from PNAC in the mice with the various gene knockouts was not due to differences in intestinal inflammation and intestinal permeability. We exposed WT mice and the respective knockout mice (i.e., *I11r*$^{-/-}$, *Casp1*$^{-/-}$, and *Ccr2*$^{-/-}$) to DSS treatment (DSS in drinking water for 4 days, followed by 2 days of chow, which represents the time point at which PN infusion would begin) and compared the colon phenotype using several validated parameters: first we visualized colitis clinically by observing the occurrence of bloody stools, which was similar in all DSS-treated mice, regardless of genotype (Supplementary Fig. 3A). Second, measurement of colon length (as an indicator of colitis) showed significantly shorter colons in all DSS-treated mice, regardless of genotype (Supplementary Fig. 3B, C). Third, colon histology (hematoxylin and eosin and F4/80 stain) showed similar severe acute inflammation of the cecum characterized by mucosal and submucosal inflammatory infiltrates, submucosal edema and expansion of lymphatics, which did not differ among mice with different genotypes treated with DSS. Specifically, there was no reduction in colon inflammation in any of the knockout mice compared to WT DSS-treated mice (Supplementary Fig. 3D). Finally, intestinal permeability was measured after gavaging DSS-treated mice with fluorescein isothiocyanate (FITC)-dextran and determining FITC-dextran concentrations in portal vein serum. We found significantly increased but similar portal vein FITC-dextran concentrations (reflecting similar increased intestinal permeability) in all DSS-treated mice, regardless of genotype, compared to non-treated mice (Supplementary Fig. 3E). Thus, the knockout mice used in these DSS/PN experiments had similar severity of colitis and increased intestinal permeability compared to the WT DSS/PN mice, discounting the possibility that differences in intestinal responses to DSS accounted for the marked differences in cholestasis and hepatic gene expression among genotypes observed in our experiments.

**IL-1β-NF-κB reduces FXR binding to target gene promoters**. We next sought to determine the mechanism by which IL-1β suppressed hepatocyte *Abcb11* and *Abcc2* mRNA expression in this model. As shown in Fig. 1, reduced *Abcb11* and *Abcc2* expression in WT DSS/PN14d mice was associated with reduced expression of the gene encoding FXR (*Nr1h4*), which induces transcription of these two canalicular transporters[36,37] and

reduced nuclear protein levels of FXR relative to chow controls. In contrast, *Il1r*$^{-/-}$ and *Ccr2*$^{-/-}$ DSS/PN14d mice had increased hepatic *Nr1h4* transcription relative to WT DSS/PN mice (Fig. 4a) and increased FXR protein in the nucleus (Supplementary Fig. 4A). Intraperitoneal injection of LPS into WT mice suppressed hepatic nuclear FXR protein levels and *NR1h4* mRNA expression. Similarly, i.p. injection of IL-1β (200 ng/mouse, for 4 h) in WT mice suppressed hepatic *Nr1h4* transcription in vivo and in vitro exposure of HuH7 cells and primary mouse hepatocytes to IL-1β (10 ng/ml, for 4 h) suppressed NR1H4/*Nr1h4* mRNA levels and nuclear FXR protein levels (Fig. 4b, Supplementary Fig. 4B, C). *Nr1h4* mRNA suppression after i.p. injection of LPS in WT mice was attenuated in mice that had undergone macrophage ablation with Clo-lip (Fig. 4c) as well as in *Il1r*$^{-/-}$ and in *Caspase1/11*$^{-/-}$ mice (Fig. 4d). Taken together, these findings support the notion that IL-1β from LPS-activated hepatic macrophages suppressed hepatocyte *Nr1h4* expression and its translocation to the nucleus.

Based on our reported in vitro findings that NF-κB interferes with FXR binding to consensus binding sequences in the promoter of *Abcb11*[46] and considering that IL-1β is a known inducer of NF-κB signaling, we next determined the role of NF-κB downstream of IL-1β in suppressing FXR and bile transporter mRNA expression in PNAC. We first exposed HuH7 cells to recombinant IL-1β, and demonstrated nuclear translocation of the NF-κB subunits p50 and p65 at 4 h of exposure. IL-1β-induced NF-κB activation was further supported by increased NF-κB luciferase reporter activity and induction of IKBA expression, a well-known NF-κB target gene (Supplementary Fig. 4D–F). Furthermore, overexpression of NF-κB p65 in HuH7 cells suppressed *ABCC2* mRNA transcription (Supplementary Fig. 4G) and inhibition of NF-κB p50 subunit increased nuclear FXR levels in IL-1β-exposed HuH7 cells (Supplementary Fig. 4C). Pre-incubation with an NF-κB inhibitor for 1 h significantly increased expression of *ABCC2* during a 4-h incubation with IL-1β; overexpression of an NF-κB repressor also increased *ABCC2* expression in IL-1β-treated HuH7 cells (Fig. 4e–f). To determine if reduced expression of *Nr1h4* was associated with reduced FXR binding and increased NF-κB binding to known consensus binding sites within the promoter of *Abcb11* in vivo in PNAC, we performed chromatin immunoprecipitation (ChIP) on liver tissue from chow mice, WT DSS/PN14d mice, and *Il1r*$^{-/-}$ DSS/PN14d mice. These experiments showed that binding of FXR to the *Abcb11* promoter was significantly attenuated in WT mice with PNAC (DSS/PN14d) compared to chow controls, but was comparable to chow mice in *Il1r*$^{-/-}$ DSS/PN mice (which were protected from PNAC) (Fig. 4g). Moreover, binding of NF-κB p50 and p65 to the *Abcb11* promoter was significantly increased in WT mice with PNAC (DSS/PN14d) compared to chow controls and was in turn decreased in *Il1r*$^{-/-}$ DSS/PN mice relative to WT DSS/PN mice (Fig. 4h). Considering the observed reduced mRNA expression of *Nr1h4* in PNAC mice, we also performed ChIP assays to examine binding of NF-κB to the *Nr1h4* promoter. Similar to the observations at the *Abcb11* promoter, we found binding of p50 and p65 to consensus binding sites within the *Nr1h4* promoter was significantly increased in WT DSS/PN14d mice compared to chow controls and was decreased in *Il1r*$^{-/-}$ DSS/PN mice (Fig. 4h). Intriguingly, binding of FXR to the *Nr1h4* promoter was also significantly attenuated in WT DSS/PN14d WT mice compared to chow controls and was similar to chow controls in *Il1r*$^{-/-}$ DSS/PN mice (Fig. 4h), suggesting self-regulation of FXR expression in this model. In summary, IL-1β induced NFκB binding to the promoters of, and appeared to suppress mRNA expression of, both *Nr1h4* and *Abcb11* in the PNAC mice.

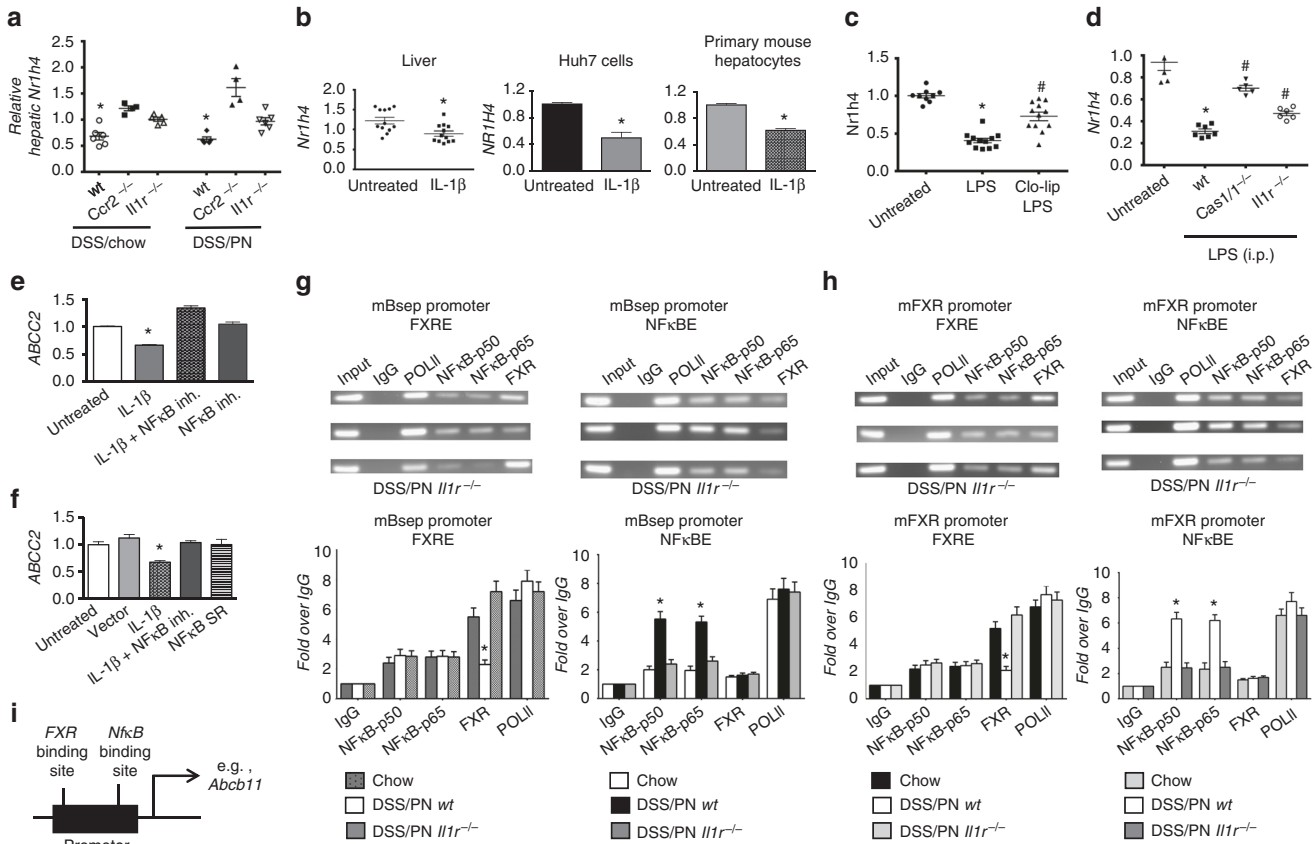

Fig. 4 IL-1β-induced NF-κB signaling suppresses *FXR* and *ABCB11*. **a** Liver mRNA for *Nr1h4* in WT, *Ccr2*[−/−], or *Il1r*[−/−] DSS/chow and DSS/PN mice. *$p <$ 0.05 vs. all other groups. One-way ANOVA and Tukey's correction. **b** Relative mRNA expression of *Nr1h4* in the liver of untreated and IL-1β- exposed (i.p. for 4 h) mice and in HuH7 cells and primary mouse hepatocytes left untreated or exposed to IL-1β for 4 h (depicting mean ± SEM of technical triplicates from one representative out of at least five experiments). *$p <$ 0.05 IL-1β vs. untreated by *t* test. **c, d** Liver mRNA for *Nr1h4* in untreated and LPS-exposed (2.5 mg/kg i.p. for 4 h) mice with and without prior i.p. treatment (16 h) with clodronate-liposomes (Clo-lip) and **c** relative to untreated and LPS exposed WT, *Casp1/11*[−/−], or *Il1r*[−/−] mice. *$p <$ 0.05 vs. untreated and #$p <$ 0.05 vs. LPS (**c**); *$p <$ 0.05 vs. untreated and #$p <$ 0.05 vs. wild type. **e, f** Relative mRNA expression of *ABCC2* in HuH7 cells left untreated or exposed to IL-1β (2 ng/ml) for 4 h with and without prior (1 h) exposure to NF-κB inhibitor (BAY-11-7082, 50 μM; designated as NF-κB in **h**) with and without transient transfection with an NF-κB repressor plasmid (NF-κB SR) (depicting mean ± SEM of technical triplicates from one representative out of at least five experiments). *$p <$ 0.05 vs. all other groups by one-way ANOVA and Tukey's correction. **g, h** Chromatin immunoprecipitation (ChIP) on liver homogenate from WT chow, WT DSS/PN, and *Il1r*[−/−]DSS/PN mice with specific antibodies for either FXR (left) or NF-κB p50 and p65 subunits (right) binding to the promotor of mouse *Abcb11* (**g**) or the promotor of mouse *Nr1h4* (**h**) represented as semi-quantitative data (top) and quantitative PCR data with specific primers (bottom). **i** Exemplary schematic depicting relative distance between binding sites for FXR and NF-κB within *Abcb11* promoter

## IL-1β-NFκB suppresses sterol transporter expression in PNAC.

We have reported that PNAC was associated with a failure to transcriptionally induce *Abcg5* and *Abcg8* encoding the canalicular phytosterol transporters, sterolin1 and sterolin2[26], which was associated with elevated hepatic and serum levels of intravenous soy lipid-derived phytosterols that have been shown to interfere with FXR signaling[18,26,47]. Therefore, we hypothesized that IL-1β also mediated suppression of *Abcg5 and Abcg8* (from here on referred to as *Abcg5/8*) in PNAC mice. We first compared hepatic *Abcg5/8* expression in DSS/PN14d mice that were either WT, *Il1r*[−/−], *Ccr2*[−/−], or Anakinra treated. Hepatic *Abcg5/8* mRNA was suppressed in WT DSS/PN14d WT mice but was induced in *Il1r*[−/−] *Ccr2*[−/−], and anakinra-treated DSS/PN mice (Fig. 5a). Similar findings were observed for expression of *Nr1h3*, the gene encoding the transcription factor LXR that induces expression of *Abcg5/8*[48] (Fig. 5a). To determine the role of IL-1β and LPS in this transcriptional suppression, i.p. injection of 200 ng/mouse IL-1β or of 2.5 mg/kg LPS was given to otherwise untreated WT mice and resulted in significant transcriptional suppression of *Abcg5/8* mRNA in livers of both groups (Fig. 5b).

Deletion of hepatic macrophages by Clo-lip treatment prevented LPS-induced suppression of *Abcg5/8* mRNA in WT mice, as did use of *Il1r*[−/−] or *Caspase1/11*[−/−] mice (Fig. 5c), supporting transcriptional suppression that was promoted by macrophage-derived IL-1β. Furthermore, in vitro exposure of HuH7 cells and of primary mouse hepatocytes to IL-1β (2 ng/ml) for 4 h suppressed mRNA levels of *NR1H3/ Nr1h3* and *ABCG5/8/ Abcg5/8* (Supplementary Fig. 5A; Fig. 5d). ChIP followed by qPCR of liver tissue from WT DSS/PN14d mice showed increased binding of p50 and p65 NF-κB subunits, concomitant with reduced binding of LXR-α (to the LXR-β site), to consensus binding sites within the *Abcg5* promoter as well as *Abcg5/8* intergenic region (Fig. 5e). In contrast, in *Il1r*[−/−] DSS/PN14d mice binding of LXR was increased and that of p50 and p65 was decreased (Fig. 5e); a semi-quantitative representation is shown in Supplementary Fig. 5B. Furthermore, in vitro exposure of HuH7 cells to 2 ng/ml IL-1β reduced nuclear LXR-α protein amounts (Supplementary Fig. 5C) and in vitro ChIP assays showed reduced binding of LXR-α to the *NR1H3* and *ABCG8* promoters concomitant with increased binding of p50 and p65 subunits (Fig. 5f). Treatment of IL-1β-

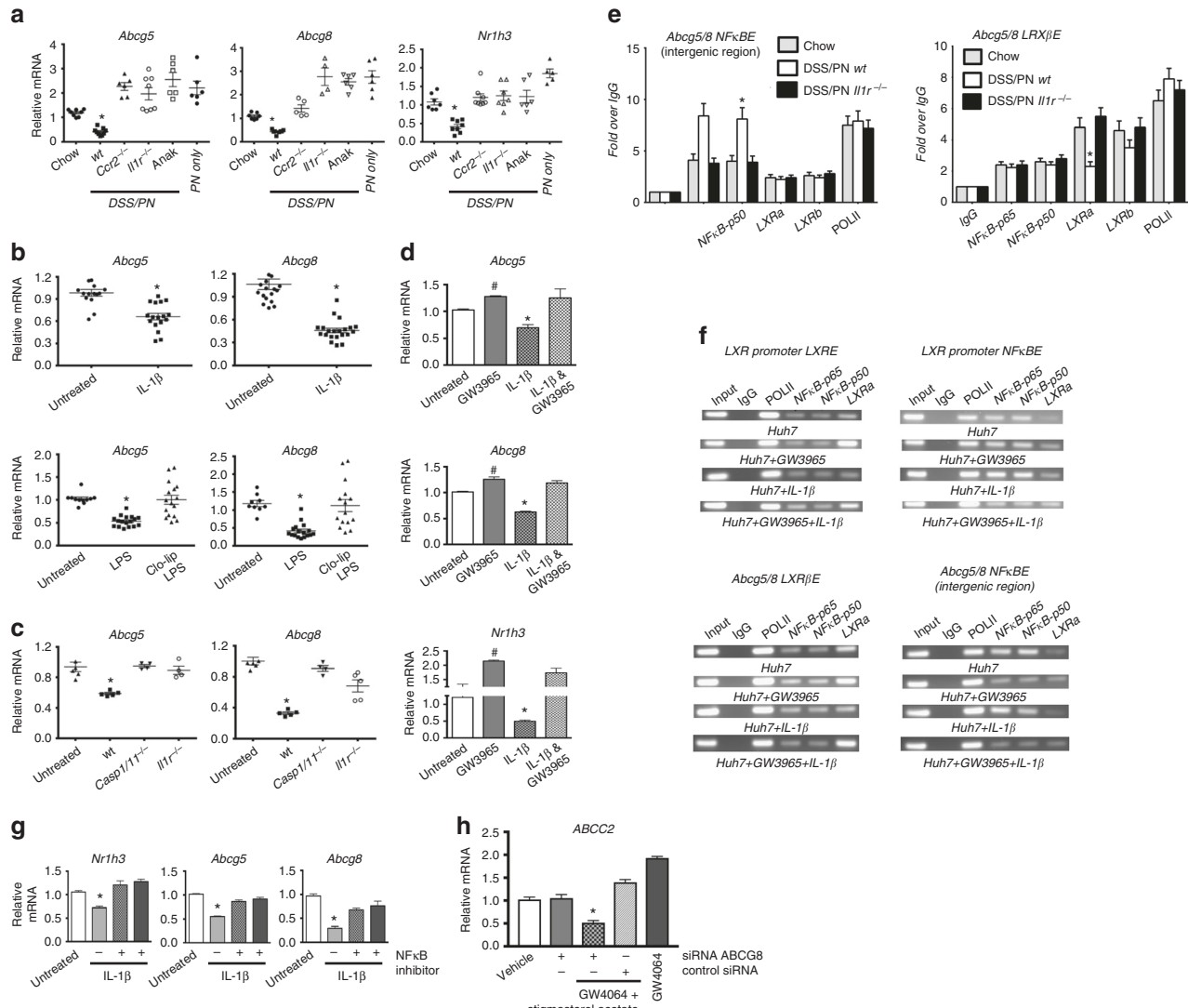

**Fig. 5** IL-1β induced NF-κB signaling suppresses *LXR* and *ABCG5/8*. **a** Liver mRNA for *Abcg5*, *Abcg8*, and *Nr1h3* in WT chow mice relative to WT DSS/PN, WT PN-only, *Ccr2^{-/-}* DSS/PN, *Il1r^{-/-}* DSS/PN, and Anakinra DSS/PN mice. **b**, **c** Liver mRNA for *Abcg5* and *Abcg8* in WT mice left untreated, or IL-1β i.p. injected (4 h) (upper panels) and in mice LPS i.p. injected (4 h), with or without prior (16 h) i.p. clodronate-liposome (Clo-lip) injection (bottom panels). *p < 0.05 vs. all other groups using *t* test (upper panels) or one-way ANOVA and Tukey's correction (lower panels). **c** Liver mRNA for *Abcg5* and *Abcg8* in WT mice left untreated, or i.p. LPS injected relative to that *Casp1/11^{-/-}* and *Il1r^{-/-}* mice. **d** Liver mRNA for *Abcg5*, *Abcg8*, and *Nr1h3* in untreated mouse primary hepatocytes relative to LXR agonist (GW3965, 2 μM, 16 h) and IL-1β (additional 4 h) treatment. **e** qPCR from LXRβ binding site within the *ABCG5/8* intergenic region and ChIP for LXRα or NF-κB p65 and p50 on liver homogenate from WT chow, WT DSS/PN, and *Il1r^{-/-}* DSS/PN mice. Data represented as fold change over IgG; POLII shown as control. **f** ChIP from HuH7 cells left untreated (HuH7), or exposed to LXR agonist (GW3965, 2 μM; HuH7 + GW3965) for 16 h with or without additional exposure to IL-1β for 4 h. Specific antibodies for either LXR (left) binding to *LXR* promoter (top left) or *ABCG8* intergenic region (bottom left) or NF-κB p50 and p65 (right) binding to the LXR promoter (top right) or *ABCG8* intergenic region (bottom right). **g** Relative mRNA in primary mouse hepatocytes for *Nr1h3*, *Abcg5*, and *Abcg8* exposed to IL-1β (2 ng/ml) with or without prior exposure (1 h) to NF-κB inhibitor (BAY-11-7082, 50 μM). **h** mRNA for *ABCC2* in HepG2 cells exposed to transfection reagent (vehicle) or control siRNA and *ABCG8* siRNA and exposure to FXR agonist GW4064 (5 μM) with and without stigmasterol acetate (20 μM) (**d**, **g**, **h** depict mean ± SEM of technical triplicates from one representative out of at least five experiments). **a**, **d**, **g** *p < 0.05 vs. all treatments using one-way ANOVA and Tukey's correction

exposed HuH7 cells with an LXR agonist increased LXR binding and suppressed NF-κB binding to the *NR1H3* and *ABCG8* promoters (Fig. 5f) and partially reversed the IL-1β-mediated suppression of *NR1H3* and *ABCG5/8* mRNA levels (Fig. 5d). In addition, in vitro overexpression of p50 or p65 subunits in HuH7 cells resulted in a dose-dependent suppression of LXR activity using an LXR reporter construct, while simultaneous overexpression of an NF-κB repressor increased LXR activity (Supplementary Fig. 5D). Finally, inhibition of NF-κB in HuH7 cells exposed to 2 ng/ml IL-1β increased mRNA expression of *Abcg5/8* and *Nr1h3* (Fig. 5g). Taken together, these data support a role for

NF-κB binding to *Nr1h3* and *Abc5/8* promoters as the mediator of the IL-1β suppression of *Abcg5/8* in PNAC mice.

Finally, we determined if reduced expression of *ABCG5/8* is permissive for phytosterols to interfere with FXR target gene expression. We used a small interfering RNA (siRNA) approach to reduce expression of *ABCG8* in HepG2 cells (resulting in ~90% reduction in mRNA for *ABCG8*, Supplementary Fig. 5E, F), which were subsequently exposed to an FXR agonist (GW4064, 5 μM; Tocris Bioscience, UK) combined with stigmasterol acetate (10 μM), a phytosterol with excellent solubility[26,47] and shown by us to promote PNAC and by others to suppress FXR

signaling[18,26,47]. The ability of GW4064 to induce transcription of its target gene *ABCC2* was significantly reduced in the presence of stigmasterol acetate when expression of *ABCG8* was suppressed by siRNA (Fig. 5h). Because of very low expression levels of *ABCB11* (Ct values consistently >36) in HuH7 and HepG2 cells, we next determined in primary mouse hepatocytes if the combination of stigmasterol acetate with IL-1β (to mediate *Abcg5/8* suppression) would inhibit the ability of GW4064 to induce *Abcc2* and also *Abcb11* transcription. Indeed, while IL-1β or stigmasterol acetate alone inhibited *Abcc2* and *Abcb11* transcription that was induced by GW4064, the greatest suppression of GW4064-induced *Abcc2* and *Abcb11* transcription was observed when IL-1β was combined with stigmasterol acetate (Supplementary Fig. 5F). Thus, reduced expression of *ABCG5/8* (either by siRNA or through IL-1β), allowing for hepatocyte accumulation of stigmasterol acetate, was associated with antagonism of FXR induction of bile transporters.

## Discussion

This study supports the hypothesis that PNAC in mice results from pro-inflammatory activation of hepatic macrophages by intestinal-derived LPS[25] and circulating phytosterols[26] with subsequent generation of IL-1β, which binds to hepatocyte IL-1 receptor and increases NF-κB translocation to the nucleus, wherein p50 and p65 bind to promoter sequences and suppress transcription of genes encoding FXR and LXR as well as FXR-regulated and LXR-regulated genes. The result is significantly reduced transcription of canalicular transporters of bile acids, bilirubin, and phytosterols causing phytosterol retention, which further antagonizes FXR induction of *Abcb11*[18,26,47], culminating in reduced biliary secretion and hepatic accumulation of bile acids which cause cholestatic hepatocyte injury (Fig. 6).

This PNAC mouse model was developed and designed in order to recapitulate the pathophysiology in the human infant and adult with PNAC, in which short bowel syndrome and other forms of intestinal injury and inflammation lead to impaired intestinal barrier function and require the patient to receive PN infusion. It is only in these patients with intestinal injury and intestinal failure that PN leads to progressive liver disease and eventual need for multivisceral transplant. Compounding the altered intestinal barrier function, the infusion of plant sterol containing lipid emulsions may activate macrophages and directly suppress bile transporters[15,25,26,29]. We therefore developed our mouse model to include both intestinal injury/increased permeability (DSS-induced) and infusion of plant sterol containing PN in order to mimic the human pathophysiology in those infants and adults that develop progressive PNAC[25,26]. In the current experiments, DSS-induced intestinal injury and inflammation and LPS absorption were not altered in the various gene knockout mice that we investigated, indicating that the demonstrated protection from PNAC in these knockout mice was likely not due to alterations in the intestinal phenotype but rather a consequence of interfering with inflammatory pathways within the liver.

Recent human studies analyzing gene expression and histology of livers from infants with intestinal failure and PNAC have further validated our mouse model by documenting very similar findings to those observed in our DSS/PN mouse model[15,29]. This includes hepatic macrophage accumulation that was associated with cholestasis and increased levels of phytosterols, increased hepatic inflammatory cytokine expression (*IL1B*, *IL6*, *TNF*) in conjunction with suppression of hepatic sterol transporters (*ABCG5/8*), and increased circulating plant sterol concentrations which coincided with suppression of canalicular bile acid transporters (*ABCB11*; BSEP) and cholestasis. Furthermore, these studies in human infants with PNAC demonstrated a relationship between increased inflammation (on liver histology and liver mRNA for *IL1B*, *IL6*, *TNF*) and decreased *ABCB11* mRNA expression[29]. These reports confirm the applicability of this PNAC mouse model to test the hypotheses put forward in the present studies.

Our in vivo experiments demonstrated that DSS and DSS/PN mice had increased accumulation of macrophages that were F4/80/CD11b double positive when analyzed by FACS, which was consistent with mRNA data showing increased expression of hepatic *Itgam* (encoding CD11b) and *Emr1* (encoding F4/80). F4/80-expressing macrophages have been suggested to mainly

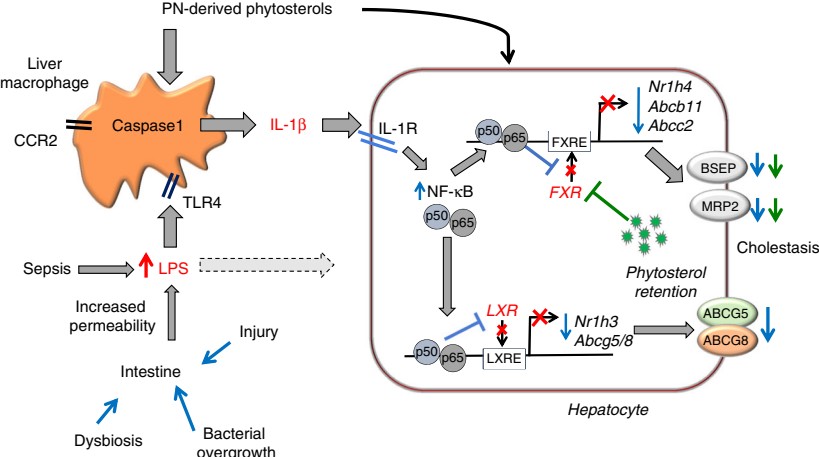

**Fig. 6** Hypothetical model of PNAC pathogenesis. Increased intestinal permeability secondary to intestinal inflammation and dysbiosis promotes increased absorption of LPS into portal vein with subsequent TLR4-mediated activation[25] and caspase-1- dependent generation of IL-1β in recruited CCR2[+] macrophages (phytosterols contribute to macrophage activation and IL-1β synthesis)[26]. Through IL-1 receptor (IL-1R) binding on hepatocytes, IL-1β subsequently activates NF-κB in hepatocytes, which interferes with the ability of LXR and FXR to bind to promoters (red crosses) and activate expression of *ABCG5/8*, *ABCB11*, and *ABCC2*, resulting in decreased expression of BSEP, MRP2, and ABCG5/8 (blue downward arrows) and phytosterol retention (green dots). Retained phytosterols antagonize FXR activity leading to additional reduction in BSEP and MRP2 expression and subsequent retention of bile acids and bilirubin (additional phytosterol effects indicated by green downward arrows) and cholestasis. In addition, direct effects of LPS on the hepatocyte can suppress FXR activity (dotted line from LPS to hepatocyte)

represent resident hepatic macrophages (Kupffer cells), while CD11b[+] macrophages appear to be recruited from circulating monocytes[32]. Therefore, macrophages with a double-positive phenotype could represent a pool of recruited macrophages that took on the resident phenotype by increasing their expression of F4/80 and maintaining expression of CD11b, as has been shown in the lung[49]. In order to functionally address if macrophage recruitment played a role in the PNAC model, we took advantage of mice genetically deficient in *Ccr2*, which have been used in other liver injury models to demonstrate a role for recruited macrophages in liver fibrosis and alcohol-induced liver injury[34,35]. We found protection against PNAC in *Ccr2*[−/−] mice, in which we demonstrated evidence of reduced macrophage accumulation by both F4/80 immunohistochemical staining and mRNA analysis. These findings are consistent with our previous report that showed increased cytokine transcription in purified CD11b[+] liver macrophages in mice with PNAC[25]. Together with increased hepatic expression of *Ccl2* mRNA (the cognate chemokine to CCR2), these data suggest an important role for recruited CD11b[+] macrophages in the pathogenesis of PNAC. More sophisticated future studies will be required to determine the degree to which recruitment of CCR2[+] macrophages contributes to the expansion of F4/80/CD11b macrophages in PNAC.

Our studies have further identified a causal role for macrophage-derived IL-1β in PNAC as evidenced by increased *Il1b* transcription in hepatic mononuclear cells and serum levels preceding the onset of cholestasis and hepatocyte injury, which can be prevented through multiple genetic and pharmacological manipulations to interrupt IL-1β signaling. Furthermore, in vitro experiments in primary mouse hepatocytes and human hepatoma cell lines HuH7 and HepG2 together with in vivo experiments demonstrated that IL-1β exposure was sufficient to suppress transcription of *Nr1h4*, *Abcb11*, *Abcc2*, and *Abcg5/8*, findings that characterize the PNAC liver. Prevention of PNAC in *Caspase1/11*-knockout mice, which are incapable of producing mature IL-1β but can produce IL-1α, verified a role for IL-1β rather than IL-1α[39,44,45]. The increased hepatic levels of cleaved caspase-1 protein (the enzyme required for processing of inactive pro-IL-1β into active mature IL-1β) in PNAC mice further support the liver as the source for IL-1β in this model. Importantly, our data demonstrate that protection from PNAC in the DSS/PN mice lacking *Ccr2*, *Caspase1/11*, or *Il1r* was not due to differences in colon inflammation or permeability. All DSS-treated mice regardless of genotype had similar timing and similar degrees of development of bloody stools, shortening of colon length, histological typhlitis, and FITC-dextran absorption into the portal vein. These data expand our previous findings that demonstrated the dual requirement of absorbed intestine-derived LPS plus intravenous administration of soy oil lipid emulsions that are unique to the pathogenesis of PNAC[25,26]. Our data suggest that IL-1β is the dominant cytokine in mediating PNAC, and that hepatic macrophage activation by gut-derived LPS and subsequent production of IL-1β might be a pathogenic pathway operative in other liver diseases involving the gut–liver axis[45]. However, it must be emphasized that the major difference in PNAC compared to other gut–liver axis disease models is the requirement for concomitant intravenous soy oil lipid emulsion administration in addition to the intestinal injury. In the PNAC mouse model, absorption of LPS from the intestine is only one of the factors leading to cholestasis, and importantly, alone is insufficient in this model to cause PNAC. DSS treatment alone to WT mice, which resulted in increased intestinal permeability and increased portal vein LPS concentrations (El Kasmi et al.[25] and Supplementary Fig. 3E), was not associated with cholestasis or liver injury in our model; however, the sequential combination of DSS followed by soy lipid emulsion containing PN was required

for the development of PNAC (El Kasmi et al.[25] and Fig. 2a,b), emphasizing the important distinction of PNAC pathogenesis from LPS/sepsis-mediated cholestasis.

Genetic and pharmacological blockade of IL-1β signaling both normalized bile transporter expression and attenuated cholestasis and hepatocyte injury in the PNAC model. Studies in humans and mice have clearly demonstrated a pivotal role for BSEP and MRP2 as canalicular transporters of bile acids and bilirubin into bile, respectively[50,51]. Disruption of expression of these transporters is pathologic as evidenced by inherited mutations of *ABCB11* and *ABCC2* which cause progressive familial intrahepatic cholestasis and Dubin–Johnson disease, respectively[50–55]. Elevated serum bile acids and bilirubin in the face of transcriptional suppression of *Abcb11* and *Abcc2* therefore strongly implicate insufficient expression of BSEP and MRP2 in the pathogenesis of cholestasis in this model. Moreover, recent human studies analyzing gene expression and histology of livers from infants with intestinal failure and PNAC have documented very similar findings to those observed in our DSS/PN mouse model[15,25,26,29], including the roles of macrophages, phytosterol levels, FXR and LXR signaling, and canalicular transporters.

Mechanistically, IL-1β appears to mediate cholestasis in PNAC through hepatocellular NF-κB signaling, which interferes with the ability of FXR to bind to and transactivate *Abcb11* and *Abcc2* promoters, thus reducing canalicular bile and bilirubin transport. Using ChIP in liver tissue from WT DSS/PN 14d mice, we showed that increased NF-κB binding coincided with decreased FXR binding to the *Abcb11* promoter. Conversely, mice deficient in the IL-1 receptor showed increased FXR binding and decreased NF-κB binding to the *Abcb11* promoter, which was associated with normalized *Abcb11* expression and attenuation of cholestasis. Furthermore, in vitro pharmacological inhibition of NF-κB increased the expression of *Nr1h4*, *Abcb11*, and *Abcc2*. These results support the proposed role of NF-κB as an important mediator of the cholestatic effect of IL-1β in this model and is consistent with previous reports demonstrating a role of NF-κB in suppressing FXR signaling and BSEP expression in vitro[46].

We also show that mice with PNAC had suppressed SHP mRNA, and that SHP expression was increased in *Il1r*[−/−], Anakinra-treated and *Ccr2*[−/−] DSS/PN mice that were protected from PNAC. Concomitantly, we found that *Cyp7a* expression was significantly higher in WT DSS/PN mice relative to *Il1r*[−/−], Anakinra-treated and *Ccr2*[−/−] DSS/PN mice. On a functional level, concentrations (measured by LC-MS) of the major taurine-conjugated bile acids were significantly increased in WT DSS/PN mice relative to chow mice, and again significantly reduced in *Il1r*[−/−], Anakinra-treated and *Ccr2*[−/−] DSS/PN mice. We conclude that reduced expression of SHP in DSS/PN mice (due to reduced expression and reduced transcriptional activity of FXR) results in insufficient suppression of *Cyp7a* with subsequent inadequate suppression of bile acid synthesis. This maladaptive failure to downregulate bile acid synthesis in cholestatic DSS/PN mice may lead to increased bile acid levels and exacerbate the cholestatic injury, especially in the face of reduced expression of BSEP.

In a 7-day PNAC mouse model, we reported that phytosterols, specifically stigmasterol, contained within the soy lipid emulsion component of the PN solution, were a required co-factor necessary for the development of PNAC[26]. Moreover, stigmasterol promoted hepatic macrophage activation in vivo and directly induced transcription of *Il1b* in vitro in naïve bone marrow-derived macrophages, thus contributing to the hepatic inflammatory milieu in this mouse model[26]. Stigmasterol has also been postulated to directly promote cholestasis through antagonizing FXR activity resulting in reduced *Abcb11* expression[18,47]. Phytosterols are excreted from the hepatocyte through canalicular sterolin1/2, a heterodimer that has been shown to be the principal

cholesterol and phytosterol transporter in the hepatocyte[56–61]. In the setting of intravenous delivery of phytosterols (e.g. PN infusion), hepatocyte excretion into bile becomes an important disposal pathway for circulating phytosterols[29,57–61]. In human infants with intestinal failure who receive PN and develop PNAC, hepatic *ABCB11* and *ABCG5/8* mRNA levels were found to be reduced and elevated serum and hepatic levels of phytosterols correlated with the severity of hepatic inflammation and cholestasis[15,16,19,29,62,63]. Similarly, significant hepatic and serum stigmasterol retention was shown by us in the 7-day PNAC mouse model co-incident with suppressed transcription of *Abcb11* and a failure to induce hepatic *Abcg5/8* expression[26]. Expanding on these observations, the current study found that *Abcg5/8* expression remained significantly reduced at 14 days in PNAC mice and that IL-1β and downstream NF-κB signaling were associated with transcriptional suppression of *Nr1h3* and *Abcg5/8*. ChIP and qPCR analyses confirmed decreased binding of LXR to consensus binding sites within the promoter of *Abcg5/8*, both in vivo in WT DSS/PN14d mice and in vitro in IL-1β treated HuH7 cells, findings that were reversed in *Il1r*$^{-/-}$-treated or Anakinra-treated DSS/PN mice, as well as upon blockade of NF-κB signaling in vitro. Similar to the findings with FXR, we found that the reduced LXR binding to *Nr1h3* and *Abcg5/8* promoters was associated with increased binding of NF-κB to these promoters, which was reversed in *Il1r*$^{-/-}$ DSS/PN14d mice. Importantly, in vitro inhibition of NF-κB signaling resulted in increased *NR1H3* and *ABCG5/8* expression in IL-1β-exposed HuH7 cells. Finally, siRNA targeted suppression of *Abcg5/8* in vitro or co-exposure of primary hepatocytes to IL-1β facilitated stigmasterol-mediated suppression of *Abcc2*. These findings demonstrate an important role for NF-κB in PNAC and further substantiate the proposed essential role of LXR in driving expression of *Abcg5/8*[46,48].

Taken together, these findings support a mechanism in which IL-1β acting through NF-κB suppressed nuclear *Nr1h4* and *Nr1h3* transcription, resulting in both reduced nuclear FXR and LXR protein levels and activity culminating in reduced bile acid, bilirubin, and sterol canalicular transport and subsequent retention of phytosterols and bile acids. Intriguingly, a recent study has implicated macrophage FXR signaling as a negative regulator in LPS-mediated IL-1β production in sepsis-associated cholestasis[64]. Therefore, increased concentrations of phytosterols (due to their structural similarity to bile acids) and bile acids as they occur in the PNAC model could further contribute to the macrophage IL-1β production through inhibition of macrophage FXR signaling. Future studies will be directed at interrogating the interplay of bile acids, phytosterols, and LPS in macrophage activation in PNAC.

In conclusion, we propose that pharmacological interference with specific inflammatory signals (e.g., macrophage recruitment and activation, IL-1β or NF-κB signaling) or modification of lipid emulsions to avoid or reduce phytosterol levels may have therapeutic value in the prevention and treatment of PNAC.

## Methods

**PNAC mouse model.** All animal procedures described herein were approved by IACUC of the University of Colorado Anschutz Medical Campus and all animals were treated humanely throughout the experimental procedures. The PNAC mouse model[25,26] employed C57BL/6 WT adult male mice (8 weeks old, 22–23 g body weight) (Jackson Laboratories, Bar Harbor, ME, USA) that were housed in specific pathogen-free conditions in individual metabolic cages and maintained on a 12 h light/dark cycle. To induce mild intestinal injury and increased permeability[25], mice were exposed ad libitum to 2.5% DSS (MP Biomedicals, Santa Ana, CA, USA) in the drinking water for 4 days during which time they had access to chow ad libitum (referred to as "DSS pretreatment"). After DSS pretreatment, PNAC mice underwent continuous pentobarbital anesthesia during surgical placement of a central venous catheter (CVC) (Silastic tubing, 0.012 inches internal diameter; Dow Corning, IL, USA) into the right jugular vein while the proximal end was tunneled

subcutaneously and exited between the shoulder plates. While still under anesthesia, mice were subsequently placed in a rubber mouse harness (Instech Laboratories, Plymouth Meeting, PA, USA) and the CVC was threaded through a swivel apparatus and connected to an infusion pump (Harvard Apparatus, Holliston, MA, USA). Mice were recovered from anesthesia on a heating pad under continuous supervision and allowed to further recovery from surgery for 24 h with intravenous (i.v.) normal saline (NS) infusion at a rate of 0.23 ml/h and were given ad libitum access to chow and water. After 24 h chow was removed, access to water was continued and mice were continuously infused for 14 days with PN (DSS/PN mice) at a rate of 0.29 ml/h providing a caloric intake of 8.4 kcal/day and soybean lipid emulsion (Intralipid$^R$, Fresenius, Bad Homburg, Germany) dose of 5 g/kg/day[25,26]. Mice were randomly assigned to each of several groups. A subgroup of DSS-pretreated mice was infused with NS for 14 days instead of PN (DSS/NS mice). Other mice (PN-only mice) were not DSS pretreated but underwent CVC placement and received PN in the same manner as DSS/PN mice. DSS-pretreated mice that did not have a CVC placed were given free access to chow and water for 14 days (DSS/chow mice). All PN infused mice had access to water ad libitum but not to chow during the PN infusion period. Unmanipulated control mice (referred to as chow mice) had free access to chow and water for a period of 19 days. In some experiments DSS/PN, DSS/chow, and chow treatments were given to *Ccr2*$^{-/-}$, *Il1r*$^{-/-}$, or *Casp1/11*$^{-/-}$ mice on the C57BL/6 background (Jackson Laboratories, Bar Harbor, ME, USA). Pharmacological blockade with the IL-1 receptor antagonist was achieved by administration of Anakinra (Kineret) 25 mg/kg body weight[45] formulated in NS and infused through the CVC on days 3 and 10 of PN infusion in DSS/PN mice. Anakinra infusion was limited to 1 h, during which PN infusion was paused. On the day of sacrifice, mice were anesthetized with i.p. pentobarbital and 100–400 µl of blood was collected from the retro-orbital plexus and liver was removed, one piece placed in formalin for 4 h followed by placement in 70% alcohol and another piece of liver was snap frozen in liquid nitrogen, coded, and subsequently stored at −80 °C until analyzed. From all mice similar lobes of liver were placed in formalin while similar lobes were used for snap freezing to ensure comparability. Coded serum samples were analyzed by the University of Colorado Hospital Clinical Chemistry Laboratory for AST, ALT, and total bilirubin levels. Total serum bile acids were analyzed in coded specimens using a Total Bile Acid Detection Kit (Diazyme Laboratories, Poway, CA, USA) according to the manufacturer's instructions. All animal procedures were approved by the Institutional Animal Care and Use Committee.

**In vivo LPS and IL-1β exposure.** C57BL/6 WT adult male mice (8 weeks old, 22–23 g body weight, identical to the criteria used for PN mice above) (Jackson Laboratories, Bar Harbor, ME, USA) were injected i.p. with 2.5 mg/kg LPS (from *Escherichia coli* 0111:B4; Sigma Aldrich, St. Louis, MO, USA) or with 200 ng/mouse recombinant mouse IL-1β (BD Biosciences, San Jose, CA, USA) before sacrifice 4 h later. After sacrifice, liver specimens were placed in formalin and snap frozen in liquid nitrogen, coded, and stored at −80 °C before RNA extraction.

**In vitro cell experiments.** Primary mouse hepatocytes were isolated from C57BL/6 WT adult male mice (8 weeks old, 22–23 g body weight) (Jackson Laboratories, Bar Harbor, ME, USA) using retrograde collagenase (Worthington Biochemical, Lakewood, NJ, USA) perfusion technique through the right atrium. Hepatocytes were washed three times and ultimately resuspended in KRH buffer (115 mmol/L NaCl, 5 mmol/L KCl, 1 mmol/L KH$_2$PO$_4$, 1.2 mmol/L MgSO$_4$, and 25 mmol/L sodium HEPES), pH 7.4, supplemented with 0.2% bovine serum albumin (Sigma Aldrich, St. Louis, MO, USA). Initial cell viability as determined by trypan blue exclusion, and was >90%[65]. Primary hepatocytes were rested overnight after isolation and stimulated the next day as indicated in the figure legends and text for a period of 4–24 h. The human hepatocyte cell line HuH7 was cultured in RPMI-1640 medium (Thermofisher, Waltham, MA, USA) with 10% fetal bovine serum (FBS) (performance FBS with low endotoxin; Thermo Fisher, Waltham, MA, USA) and 1% penicillin and 1% streptomycin; Thermo Fisher, Waltham, MA, USA). All cells were grown in 5% CO$_2$ in a humidified incubator maintained at 37 °C. The cell lines were obtained from the American Tissue Culture Collection (Manassas, VA, USA)[46]. Cells were tested for mycoplasma to ensure absence of contamination each 6 months using MycoAlert Mycoplasma Detection Kit (Lonza, Walkersville, MD, USA).

HuH7 cells were stimulated with LPS (100ng/ml; *Salmonella typhimurium*, Sigma Aldrich, St. Louis, MO, USA) or recombinant human IL-1β (2–10 ng/ml; BD Biosciences, San Jose, CA, USA) for 4 h in the presence or absence of NF-κB inhibitor which was pre-incubated for 1 h (50 µM, BAY-11-7085; Cayman Chemicals, Ann Arbor, MI, USA) or in the presence or absence of an LXR agonist which was pre-incubated for 1 h (10 µM; T0901317; Tocris, Minneapolis, MN, USA).

HepG2 cells (ATCC, Manassas, VA, USA) were transfected with control siRNA or siRNA targeting *Abcg8* using DharmaFECT-Duo (Dharmacon, Lafayette, CO, USA) according to the manufacturer's protocol. Control siRNA or siRNA targeting *Abcg8* were both designed, synthesized, and prepared by Dharmacon. HepG2 cells were cultured for 48 h before treatment with 5 µM of the FXR agonist GW4064 (Tocris, Minneapolis, MN, USA) and/or 10 µM stigmasterol acetate (Steraloids, Newport, RI, USA) for 20 h in same media as HuH7 cells (but containing 0.25%

serum). HuH7 cells and HepG2 cells were used as cell lines representing human hepatocytes.

**Immunoblot.** Immunoblotting of cytosolic and nuclear extracts (using NE-PER™ Nuclear and Cytoplasmic Extraction Reagents according to the manufacturer's descriptions; Thermo Fisher) was performed using antibodies for FXR-α (Abbiotec, cat# 252165, polyclonal rabbit, 1:1000), LXR-α (Thermo Fisher Scientific, cat# PA1-332, polyclonal rabbit, 1:500) and NFκB p50 (Abcam, cat# ab7971, polyclonal rabbit, 1:1000), and p65 (Cell Signaling, cat# 8242, monoclonal rabbit, 1:1000) subunits[46]. Detection of calnexin (Enzo Life Sciences, cat# ADI-SPA-860, polyclonal rabbit, 1:10,000) and lamin B (Santa Cruz Biotechnology, cat# sc-6217, polyclonal goat, 1:1000) were used to detect potential contamination of cytoplasmic and nuclear extracts, respectively.

Immunoblotting of total protein extracts from liver homogenate was performed using M-PER™ Extraction Reagents according to the manufacturer's descriptions; Thermo Fisher) using an antibody specific for procaspase-1 and cleaved caspase-1 (Millipore, cat# 06-503, polyclonal rabbit, 1:1000). Anti-GRB2 was used as a loading control (Cell Signaling, cat# 3972, 1:1000).

The secondary antibodies used for NF-κB subunits p50 and p65, calnexin, and lamin B are from LiCor Biosciences (Donkey anti-Rabbit, cat# 926-68073, 1:5000; Donkey anti-Goat, cat# 926-68074, 1:5000). The secondary antibody used for all other blots was from Cell Signaling (anti-rabbit HRP, cat# 7074S, 1:5000). Uncropped scans of the blots depicted in the main body of manuscript are shown in Supplementary Fig. 7.

**Immunohistochemistry.** Liver and colon tissues were removed at sacrifice, and then formalin-fixed, paraffin-embedded, and prepared for histology. Slides were stained with hematoxylin and eosin and the pan macrophage marker F4/80 (clone BM8; BMA Biomedicals, Rheinstrasse, Switzerland) at a dilution of 1:200. Slides were treated with a Proteinase K (Invitrogen, cat# 25530-049) antigen retrieval technique at 20 μg/ml in Tris-EDTA CaCl buffer, pH 8, in a 37 °C water bath for 6 min. Standard immunohistological protocols were used thereafter. Coded slides were examined and scored in a blinded fashion by a veterinary pathologist (L.J.).

**IL-1β and IL-1 receptor antagonist ELISA.** The mouse IL-1βIL-1F2 Quantikine ELISA Kit (R&D Systems, Minneapolis, MN, USA) and the human IL-1RA Quantikine ELISA Kit (R&D Systems, Minneapolis, MN, USA) were used according to the manufacturer's instructions to detect serum mouse IL-1β and serum IL-1RA, respectively.

**RNA isolation and quantitative gene expression analysis.** Liver or cellular RNA was extracted using TRIzol® (Invitrogen, Carlsbad, CA, USA) according to manufacturer's instructions. All RNA samples were DNAse treated (Ambion/Applied Biosystems, Austin, TX, USA) and reverse transcribed with iScript (Bio-Rad, Hercules, CA, USA). Gene expression was analyzed by quantitative reverse transcription PCR (qRT-PCR) analysis on an Applied Biosystems 7300 or 7500 cycler and normalized to endogenous control gene *Hprt1* (encoding hypoxanthine phosphoribosyltransferase1). Gene expression is depicted as relative mRNA amounts (relative quantities (RQ)) after normalization to expression of endogenous control gene *Hprt1* calculated using delta/delta Ct method using the software provided by Applied Biosystems that comes installed on the7300 or 7500 cyclers. WT untreated mice or WT chow mice were used as normalizer (RQ set to 1) for comparison of gene expression in WT mice and chow mice from respective gene-targeted mice were used as normalizers (RQ set to 1) for comparison of gene expression in gene-targeted mice. For in vitro experiments, gene expression is depicted as relative mRNA amounts after normalization to expression of endogenous control gene *Hprt1* (mouse cells) and *HPRT1* (human cell lines) calculated using delta/delta Ct method as described above and untreated cells were used as normalizer (set to 1) for comparison of gene expression. PCRs were performed in technical triplicate for cDNA from each animal and each target gene and *Hprt1* shown are averages of the individual means obtained from triplicates. For all gene expression experiments, samples were analyzed in triplicate using TaqMan probes, which were designed, tested, optimized, and distributed by Applied Biosystems (commercially available standardized "TaqMan gene expression assays"). All qRT-PCR reagents and supplies, including TaqMan mastermix were from Applied Biosystems, and all procedures were carried out according to protocols and using software from Applied Biosytems. To verify the fidelity of gene expression results in this model, for some target genes (*Abcg8*, *Abcb11*, *Nr1h3*, *Nr1h4*), mRNA expression was determined using an additional housekeeping gene (*Ee2f*) side by side with *Hprt1* using cDNA from RNA obtained from liver samples from a representative group ($n = 3$–5) of chow and DSS/PN mice with WT, *Il1r⁻/⁻* or *Ccr2⁻/⁻* genotype in order (Supplementary Fig. 6).

**Chromatin immunoprecipitation.** ChIP was performed on liver samples that were immediately flash frozen in liquid nitrogen before processing or on freshly obtained cell lysates from HuH7 cells[46]. ChIP was done using specifc antibodies for FXR (Abbiotech, cat# 252165, rabbit polyclonal), LXRα (Abcam, cat# ab41902, rabbit polyclonal), LXRβ (Thermo Fisher Scientific, cat# PA1-333, rabbit polyclonal), NF-κB p50 (Cell Signaling, cat# 13586, rabbit polyclonal) and NF-κB p65

(Abcam, cat# ab16502, rabbit polyclonal) and amplification of promoter sequences from the *Abcb11*, *Nr1h3*, *Abcg5/8*, and *Nr1h4* genes using specific primer sets and subsequent PCR; primer sequences will be provided upon request[46]. ChIP assays were performed by a combination of protocols and according to the manufacturer's instructions using EZ ChIP/MagnaChIP G Kit from EMD/Millipore (Billerica, MA, USA)[46]. Cells were harvested after fixation with 1% formaldehyde. After cell lysis and sonication, the fragmented DNA was diluted in ChIP dilution buffer and pre-adsorbed with protein G sepharose/salmon sperm DNA (EMD/Millipore, Billerica, MA, USA) for 1 h at 4 °C. Then 5% of the chromatin was removed and saved as input. It was then incubated overnight at 4 °C with 3–5 μg of the appropriate antibodies or normal mouse IgG (Product 12-371 from EMD/Millipore, Billerica, MA, USA). Antibody–chromatin complexes were captured by incubation with protein G sepharose and centrifuged. Reversal of protein cross-linking and proteinase K digestion, followed by purification of the DNA, was then achieved. An aliquot of the DNA (2 μl) was used in a PCR (standard and quantitative) reaction using specific primers flanking the FXRE, LXRE, or NF-κBE of human *ABCG5/8* and *LXR* promoters. Primers flanking a site distant from the FXRE, LXRE, and NF-κB sites were used as negative controls. PCR products were run in a 2% agarose gel and stained with ethidium bromide to confirm the amplicon size.

The method for in vivo ChIP analysis of liver has been modified from protocols for cells[46]. Mouse livers were sliced to small pieces and then incubated with 1% formaldehyde to crosslink proteins to genomic DNA in cells. Following this incubation, excess formaldehyde was quenched by incubation with glycine. Chromatin was prepared from cross-linked livers after isolation of nuclei. Chromatin was sonicated with the appropriate power setting to shear DNA to approximately ~500 bp fragments for use in ChIP. Uncropped scans of the gels depicted in main body of manuscript are shown in Supplementary Fig. 7.

**UHPLC-MS quantification of bile acids in serum and tissue.** A solution of stable isotope-labeled bile acid standards (50 nM each, all 2,2,4,4-deuterated, Cambridge Isotope Laboratories, Tewksbury, MA, USA) was prepared in Optima methanol (Fisher Scientific, Pittsburgh, PA, USA): cholic acid, chenodeoxycholic acid, deoxycholic acid, glycochenodeoxycholic acid, glycocholic acid, taurochenodeoxycholic acid, taurocholic acid, tauodeoxycholic acid, taurolithocholic acid, tauroursodeoxycholic acid, and ursodeoxycholic acid. Serum samples were thawed on ice and extracted at a 1:25 dilution. Frozen tissue samples were weighed to the nearest 0.01 mg and extracted at 80 mg/mL (liver) or 40 mg/mL (intestinal content). Samples were agitated at 4 °C for 30 min followed by centrifugation at $10,000 \times g$ for 10 min at 4 °C. Supernatant aliquots from tissue extractions were diluted with the above MeOH solution to a final concentration of 20 mg/mL. Supernatants (15 μL for serum; 10 μL for tissue) were injected into a Thermo Vanquish UHPLC system (San Jose, CA) and run on a Kinetex C18 column (150 × 2.1 mm², 1.7 μm; Phenomenex, Torrance, CA, USA) at 300 μL/min. Solvents were A—5% acetonitrile, 5 mM ammonium acetate; B—95% acetonitrile, 1 mM ammonium acetate. Water, acetonitrile, and ammonium acetate were Optima grade. The gradient was 0–1 min at 17% B, 1–2 min 17–72% B, 2–7 min 72–74% B, 7–7.1 min 74–100% B, 7.1–8 min hold at 100% B, 8–8.1 min return to 17% B, and 8.1–10 min hold at 17% B. The autosampler was held at 7 °C and the column compartment at 25 °C. The UHPLC system was coupled online with a Thermo Q Exactive mass spectrometer (Bremen, Germany), scanning at 70,000 resolution in the 350–550 $m/z$ range in negative ion mode. Negative electrospray ionization was achieved with 4 kV spray voltage, 25 sheath gas, and 5 auxiliary gas. Data were analyzed using Maven (Princeton, NJ, USA). Absolute quantitation was performed by measuring the ratios of labeled and endogenous bile acids.

**Intrahepatic mononuclear cell isolation.** Buffers and solutions were kept at 37 °C in a water bath. Solution 1: 50 ml EBSS (Gibco, Lafayette, CO, USA), 500 μl HEPES (Sigma Aldrich, St. Louis, MO, USA), 9.5 mg EGTA (Sigma Aldrich, St. Louis, MO, USA). Solution 2: 50 ml liver perfusion medium (Gibco, Lafayette, CO, USA) and solution 3: 50 ml of liver digestion medium (Gibco, Lafayette, CO, USA). Mice were kept on a heating pad and anesthetized with pentobarbital before performing a midline laparotomy that extended to the ribcage. Then, the portal vein was identified and revealed and cannulated with a 24 gauge catheter that was connected to plastic tubing for perfusion. Solution 1 was perfused at a rate of 2 ml/min until the liver became opaque (approximately after 3 min of perfusion). Then, the inferior vena cava was incised (to allow unimpaired outflow) before performing a sternotomy and clamping the superior vena cava. Subsequently, solutions 2 and 3 were perfused for 6 min each. Liver was then carefully removed from the mouse and placed into a chilled petri dish on ice with 5 ml of Williams medium (Thermofisher, Waltham, MA, USA) containing high-performance FBS (Thermofisher, Waltham, MA, USA) and 1% penicillin and streptomycin (Thermofisher, Waltham, MA, USA). Liver capsule was incised, tissue gently swirled to liberate cells, and subsequently passed over a 70 μm strainer into a chilled 50 ml tube, topped off with Williams medium, and hepatocytes were pelleted by gravity for 30 min. Supernatant (containing non-hepatocytes) was transferred to a fresh tube and centrifuged for 5 min at $200 \times g$. Cell pellet was washed once in phosphate-buffered saline (PBS) and then resuspendend in 1.6 ml of PBS and cell suspension mixed into 2.4 ml of chilled 40% histodenz (Sigma Aldrich, St. Louis, MO, USA). One milliliter of cold PBS was overlayed on top of the cell mixture and centrifuged at 2563 rpm for 20 min with the brake off. At the resulting interface of the PBS to

gradient mononuclear cells were recovered and transferred to a tube containing 15 ml of PBS and pelleted at 1200 rpm for 5 min before further use for flow cytometry or RNA extraction.

**Flow cytometry**. IHMCs were isolated from the liver as described above, washed, pelleted, and resuspended in staining buffer, PBS with 1% Hyclone FBS (GE Healthcare Life Sciences). An aliquot was removed, stained with trypan blue, and counted via a hemocytometer[66]. All antibodies were purchased from BD Biosciences (San Jose, CA, USA) and used at a diluton of 1:100. In brief, FcγR was first blocked with anti-CD16/CD32 antibody at 4 °C for 10 min. Cells were then stained for 30 min at 4 °C with the following antibodies: Ly6G-BV421 (Clone 1A8), CD45R-BV421 (Clone RA3-6B2), CD3-BV421 (Clone 17A2), CD45-PE (Clone 30-F11), F4/80-AF488 (Clone T45-2342), and CD11b-PE/Cy7 (Clone M1/70). All sort tubes were pre-coated with FBS and contained ice-chilled sort buffer (PBS, 1 mM EDTA, 25 mM HEPES, pH 7.3, and 50% FBS). Flow-assisted cell sorting was conducted by staff at the University of Colorado Cancer Center Flow Cytometry Core Facility and were performed on an XDP-100 device (Beckman Coulter). The sorting strategy excluded debris and cell doublets using light scatter (forward scatter and side scatter) while dead cells were excluded by DAPI$^+$ (4′,6-diamidino-2-phenylindole) staining (1 mg/ml). Fluorescence minus one controls were used to determine gating areas. Neutrophils, B cells, and T cells were excluded in the Dump$^+$ gate (Supplementary Fig. 1B). Dump$^-$ gate was then selected and CD45$^+$ population was then segregated into monocytes (Mono: Dump$^-$, CD45$^+$, F4/80$^{Lo}$, CD11b$^{Hi}$), recruited macrophages (RecMO: Dump$^-$, CD45$^+$, F4/80$^{Hi}$, CD11b$^{Hi}$), and Kupffer cells (KCs: Dump$^-$, CD45$^+$, F4/80$^{Hi}$, CD11b$^{Lo}$). All data analysis was performed using Kaluza 1.5 (Beckman Coulter).

**Statistical analysis**. Gene expression was determined in triplicate for each mouse in qRT-PCR assays. The average of the triplicate from each mouse was used to generate the mean±standard error of mean for gene expression in each treatment group; the number of animals in each group was generally 5–10 and are provided in the text, figures, and/or legends. For cell experiments, gene expression was determined in triplicate and results from one representative out of at least three experiments are shown. One-way analysis of variance and Tukey's correction for multiple comparisons were used to determine statistical significance if more than two groups were analyzed, otherwise Student's $t$test was used. If not otherwise specified, a $P$ value <0.05 was considered statistically significant.

**Data availability**. The data generated during and/or analyzed during the current study are available from the corresponding authors on reasonable request.

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

## Acknowledgements

This project was supported in part by NIH grants UL1TR001082 and T32 DK067009 (awarded to R.J.S.) and RO1 DK84434 (awarded to F.J.S.). The contents are the authors' sole responsibility and do not necessarily represent official NIH views.

## Author contributions

Principal design of the study, conception of study, analysis and interpretation of data, writing of manuscript: K.C.E., R.J.S.; contribution to design of experiments and interpretation of data and writing of manuscript: C.W., F.J.S., P.M.V., E.N.-G. Data generation and experimental procedures: P.M.V., A.L.A., M.W.D., N.B., S.G., C.J.W., S.M., A.L.A.; C.D., S.A.F., A.D., J.R.), histopathological analysis and interpretation: L.J.

## Additional information

**Competing interests:** The authors declare no competing interests.

