## [Peer Review File · Nature Communications]

Reviewers' comments:

Reviewer #1 (Remarks to the Author):

El Kasmi present a nice series of experiments on the role of CCR2, Caspase 1/11, IL-1 receptor and the therapeutic effects of pharmacological IL-1 antagonism in the DSS+PN infusion model in mice to mimic PNAC in children. For that aim authors used PN infusion for 14d in DSS treated (2.5.%) mice and used challenged respective KO mice or used IL-1 blocking ABs. In essence they show that genetic loss or blockage of CCR, caspase 1/11, and IL-1 ameliorates liver injury and repression of transporter mRNA levels and expression of FXR and LXR.

Major points:

1. The construct validity of the used model is questionable. This should be discussed in more detail.
2. Repression of hepatic transporter expression (transcriptionally and posttranscriptionally) in cholestatic liver diseases especially in inflammatory induced cholestasis is a well known fact since the late 90s (Karpen S, Trauner M, Boyer J, Mueller, Keppler D) - therefore the findings of the current study are somehow incremental in its nature - authors should more specifically state what is actually new on their current data.
3. Authors should give detailed data on colon injury following DSS challenge. Since they used total body KOs or blocking ABs the effects on liver injury may secondarily be related to reduced colitis? Consequently one would either do colonoscopy in the different mouse strain during the experiments or alternatively just do the DSS challenge in the different strains and compare the colon phenotype.

Reviewer #2 (Remarks to the Author):

The authors have developed a robust mouse model of parenteral nutrition-associated cholestasis (PNAC). Their preceding publications have shown that PNAC in their mouse model requires two elements: an overload of phytosterols largely derived from soy components in the parenteral nutrition, and an intestinal injury achieved by a short pretreatment with dextran sulfate sodium in the drinking water. This paper represents a dissection of the mechanistic interplay between these elements using multiple knockout mouse models and pharmacological interventions. Beginning with the basic identification of recruited macrophages as a component of the pathogenic response, the authors ultimately drill down to the demonstration that NFkB competes for binding with Lxr and Fxr in the regulatory regions of genes encoding key canalicular transporters (Abcb11, Abcc2, and Abcg5/8). The down-regulation of these transporters, which might otherwise attenuate the phytosterol and/or bile acid accumulation, thereby contributes to the pathogenesis of PNAC. Since bile acid accumulation has long been recognized to have a cross-regulatory pro-inflammatory effect, this suggests the possibility that in PNAC a feed-forward loop exists whereby gut-liver inflammatory signals such as LPS trigger an initial cholestatic event, the resulting local increase in retained bile acid amplifies the inflammation, further promoting or establishing long-term cholestasis. Importantly, the demonstration that the PNAC can be completely attenuated in mice using an existing FDA-approved drug that blocks IL-1b, has potentially immediate translational implications.

Major criticisms

All of the gene expression throughout the paper is interpreted by normalization to a single reference gene (Hprt1). This is not fully appropriate given the variety of models, conditions, differences in cell population (due to inflammatory infiltrate), and physiological states of the tissue and cells examined. MIQE guidelines recommend normalizing the geometric mean of at least three reference genes, whose suitability should be determined empirically for the system under study. Indeed, a recent study demonstrated that Hprt levels are significantly altered in response to DSS

treatment (Eissa et al., PLoS One 2016, PMID: PMC4886971). All least three additional reference genes should be measured in the samples presented, and the data reanalyzed (expression level of the reference genes can be measured in the same samples without necessarily re-measuring all of the experimental genes) relative to the geometric mean of the new normalizers. Failing this, the gene expression data cannot be interpreted with any degree of confidence.

Minor criticisms

Since the role of FXR in down-regulating CYP7A1 by increasing the expression of SHP (NROB2) is a well-known and important component of feedback regulation of bile acid synthesis, particularly under conditions where bile acid levels are high in the liver, but strikingly there was no mention of this. It's possible that Nr0b2 levels are decreased in the PNAC livers due to the decrease in FXR, and that this increases Cyp7a1, exacerbating the cholestasis. This would be trivial to test in the existing samples and would be a nice addition.

Fig 1C Data described as relative mRNA, but it's not clear what they are relative to (chow, WT?). It would seem more sensible to normalize the RQs to DSS/PN WT, at least in this panel.

Line 312: Fig 4,E F text and axis label refer to ABCC2, but the figure legend refers to NR1H4. Please correct the discrepancy.

Line 318: The use of the word "restored" with respect to the DSS/PN II1r -/- mice may not be appropriate, since they are protected from PNAC from the outset. I suggest changing it to something more neutral like "Fxr binding to the Abcb11 promoter in DSS/PN II1r -/- mice was comparable to that of chow-fed WT controls".

Multiple instances throughout: "Tukey" misspelled as "Tuckey".

Fig 4, Fig 5 schematics of the relevant promoters and response elements should be provided.

Line 457 should read "sterolin" not "sertolin".

Figure 5F should be presented as the qPCR result, not the qualitative gel images. I suggest cutting the gel image to supplemental material from both Fig 4 and Fig 5.

Line 611 Supplier, product number, and (if possible) lot number should be included for the ChIP antisera.

Line 613 primer sequences must be included as a supplemental table.

Line 637 What constitutes a "reasonable request"?

General comments: the presence of several convoluted run-on sentences made parts of the text unnecessarily challenging to read. The figures could be improved by increased attention to detail with respect to stylistic consistency: consistent line weights, axis number formats, axis alignment, axis label rotation.

Reviewer #3 (Remarks to the Author):

In their manuscript El Kasmi et al. investigate the role of macrophage mediated IL1-beta/NFkB signaling during parenteral nutrition associated cholestasis (PNAC). The current manuscript builds up on their recently published in-vivo model of PNAC (El Kasmi et al., Science Translational Medicine 2013). These earlier data gave compelling mechanistic evidence that phytosterol

promotes liver injury in PNAC. Their earlier work further demonstrated that gut-derived microbiota associated molecular patterns (MAMPs) and TLR4 activation due to intestinal injury represent essential prerequisites for PNAC. In their current manuscript they aim to further elucidate the cellular and molecular mechanisms involved in PNAC. To this end, they employ a combination of genetic knock-out mouse models and pharmacological approaches to highlight the important function of hepatic macrophages and IL1-beta. Although I consider the current data relevant to extend our knowledge about the cellular and molecular mechanism of PNAC, I have some major concerns with this manuscript.

1) In figure 2 the authors demonstrate the functional impact of macrophage derived IL1-beta on expression of hepatic bile acid transporters. The presented data argues that IL1-beta is the predominant inflammatory cytokine inducing PNAC. However, the downstream target NFkB is activated by a variety of different inflammatory mediators. Earlier data showed that Tnf-alpha is an important driver of endotoxin-induced cholestasis. It is essential to include functional Tnf-alpha data to demonstrate that IL1-beta is really the only mediator of this effect in PNAC to better understand the present data compared with earlier published data. Further, it is important to delineate the molecular mechanism of inflammatory macrophage activation. The current data show that cleavage of pro-IL1beta by Casepase1/11 is important. Based on the current data, it seems likely that the Nlrp3 inflammasome mediates PNAC. Gene expression analyses and staining of Nlrp3-sensor molecule are required to address this question.

2) Previous reports by the group have already pointed to the role of hepatic macrophages in PNAC. In the first part of their manuscript they further aim to define those hepatic macrophages. To this end, mRNA expression analyses of Ccl2, Ccr2, Itgam and Emr1 were performed and found to be significantly increased compared to controls. Indeed, the Ccr2-Ccl2 axis plays an essential role in macrophage recruitment upon liver damage. However, gene expression analyses by far do not suffice to thoroughly phenotype the hepatic immunological response upon PNAC injury. To address this issue, I would highly recommend multicolor flow cytometry analyses of whole liver homogenate using a staining panel to differentiate granulocytes, Kupffer cells, different subsets of monocyte derived macrophages (MoMF) and monocytes. Additional analyses of macrophage activation state would be interesting.

3) Figure 4C shows that reduced Nr1h4 mRNA expression after i.p. injection of LPS in WT mice is attenuated in mice that had undergone treatment with clondronate liposomes. In contrast to the author's conclusion, this treatment does not deplete all hepatic macrophage subsets and is not specific for hepatic macrophages as basically all phagocytic cells are affected. It is essential to delineate the contribution of individual myeloid cell subsets (Kupffer cells vs. MoMFs vs. monocytes) using flow cytometry to understand the cellular mechanism of the PNAC associated inflammatory response. Gadolinium chloride could be used to specifically target hepatic Kupffer cells.

4) Reduced inflammatory macrophage infiltration in CCR2-/- mice should not be confirmed based on mRNA expression. Flow cytometry analyses or tissue stainings against CD45 (as pan leukocyte marker), CD11b, F4/80 and Ly6G are needed.

5) The author's conclusion - in line 189-191 - that pathways associated with macrophage activation suppressed expression of bile acid transporters can not be reached based on the presented data. Indeed, it would strengthen the manuscript to analyze the activation status of hepatic macrophages and characterize the (most likely) macrophage driven inflammatory response. However, the current data neither answers which macrophage subset is involved (Kupffer cells, different MoMF subsets, classical monocytes (Ly6Chi, CCR2+, CX3CR1low), Ly6Clow patrolling monocytes) nor does it further characterize the activation state of these cells.

6) In a translational perspective, it would be valuable to confirm the proposed mechanism in

human PNAC samples (Serum II1-beta and endotoxin, tissue stainings, gene expression analyses).

Response to Reviewer #1:

We thank the Reviewer for the positive appraisal of our manuscript and for emphasizing the genetic and pharmacological value of the approach, and for suggesting clarification of important aspects of our study.

Response to major points:

1. The construct validity of the used model is questionable. This should be discussed in more detail.

Response: We thank the reviewer for this comment and we appreciate the opportunity to clarify the justification for use of this mouse model to study the mechanisms of human PNAC. This model was developed and designed in order to recapitulate the pathophysiology in the human infant and adult with PNAC, in which short bowel syndrome and other forms of intestinal injury and inflammation require the patient to receive continuous PN infusion. It is only in these patients with intestinal injury and intestinal failure that PN leads to progressive liver disease and eventual need for multivisceral transplant. Our hypothesis is that the altered barrier function in the immature intestine subjected to injury and inflammation promotes the absorption of bacterial products that may activate the hepatic innate immune system and cytokine generation which then suppress hepatocyte bile and sterol transporters. In addition, the infusion of plant sterol containing lipid emulsions further activates hepatic macrophages and may directly suppress bile transporters.

We developed our mouse model to include both intestinal injury/increased permeability (DSS-induced) and infusion of plant sterols in order to mimic the human pathophysiology in those infants and adults that develop progressive PNAC.

We would like to emphasize that while other groups have also studied animal models of TPN-induced liver injury (Dr. Burrin's group PMID: 24478031; Dr. Willing's group PMID: 27495286; Dr. Wales group PMID: 25326097; Dr. Guo's group PMID: 26564717; Dr Teitelbaum's group PMID: 19179882) in both the piglet and mice, those models have not taken into account the critical aspect of intestinal injury and increased permeability that accompanies the most severe forms of PNAC in human infants and adults. Our model, designed to combine intestinal injury and TPN, demonstrated increased intestinal permeability and absorption of intestinal bacterial products into the portal circulation, as well as the necessity for the infusion of plant sterol- containing lipid emulsions.

Verification of the applicability of our PNAC mouse model includes recent human studies analyzing gene expression and histology of livers from infants with intestinal failure and PNAC, which have documented very similar findings to those observed in our DSS/PN mouse model (Dr. Pakarinen's group: PMID: 28234635; PMID: 26962062). This includes hepatic macrophage accumulation, increased hepatic inflammatory cytokine expression (IL1b, Il6, TNF) in conjunction with suppression of hepatic sterol transporters (*ABCG5/8*) and increased circulating plant sterol concentrations, coinciding with suppression of canalicular bile acid transporters (*ABCB11*, BSEP) and cholestasis. Furthermore, these studies in human infants with PNAC demonstrated a relationship between increased inflammation on liver histology and decreased BSEP expression, which recapitulates the findings in our mouse model (**PMID: 24107776; PMID: 22120983**). Thus, when considering these recently published findings in human infants, our DSS/PN model closely resembles the human pathophysiology. Therefore, we believe that this model

should be viewed as having construct validity to test the hypotheses that we put forward. A statement describing this validation of our mouse model has now been added to the Discussion.

- 2. Repression of hepatic transporter expression (transcriptionally and post-transcriptionally) in cholestatic liver diseases especially in inflammatory induced cholestasis is a well-known fact since the late 90s (Karpen S, Trauner M, Boyer J, Mueller, Keppler D) - therefore the findings of the current study are somehow incremental in its nature - authors should more specifically state what is actually new on their current data.**

Response: We appreciate this comment, as we are aware of the observations of LPS or sepsis-induced cholestasis. However, we want to emphasize that in our PNAC mouse model, absorption of LPS from the intestine is **only one of the factors** leading to cholestasis, and importantly, alone is insufficient in this model to cause PNAC. DSS treatment alone to wt mice, which resulted in increased intestinal permeability and increased portal vein LPS concentrations (PMID: 22120983 and current data), was **not associated with cholestasis** or liver injury in our model, however the sequential combination of DSS followed by soy-lipid emulsion containing PN was required for the development of PNAC, emphasizing the distinction of PNAC pathogenesis from LPS/sepsis mediated cholestasis. Thus, the mechanisms of cholestasis in this mouse model differ significantly from those referred to by the reviewer, possibly because of the **lower levels of LPS absorption** into the portal vein in this model compared to LPS exposure after i.p. LPS injection or in models of bacterial sepsis. Based on these differences, we believe our findings are more than incremental extensions of previous work of others examining the effect of LPS on liver. We have revised our Discussion to reflect this distinction.

- 3. Authors should give detailed data on colon injury following DSS challenge. Since they used total body KOs or blocking ABs the effects on liver injury may secondarily be related to reduced colitis? Consequently, one would either do colonoscopy in the different mouse strain during the experiments or alternatively just do the DSS challenge in the different strains and compare the colon phenotype.**

Response: We agree with the reviewer that DSS induced intestinal injury and increased permeability are important aspects of this model, and that it is important to ensure that differences in colon inflammation in the different genotypes were not responsible for our findings. To address this point experimentally, we have exposed wild type mice and the respective knock-out mice (i.e. IL1R^{-/-}, Casp1^{-/-} and CCR2^{-/-}) to DSS treatment (DSS for 4 days, followed by 2 days chow, which represents the time point at which PN infusion would begin) and compared the colon phenotype using several validated parameters:

- 1) We visualized colitis clinically by observing the occurrence and timing of bloody stools, which occurred similarly in all DSS-treated mice, regardless of genotype (this is now presented as a photograph in supplementary Figure 3A).
- 2) We also measured colon length in these mice as an indicator of colitis (shorter colons represent more extensive inflammation, as shown by many other groups). Colon length was significantly reduced and was similar in all DSS-treated mice,

- regardless of genotype (this is now presented as a photograph in supplementary Figure 3B, C).
- 3) Furthermore, we examined colon histology (H&E and F4/80 stain) in these DSS-treated mice, which showed similar severe acute inflammation of the cecum, characterized by mucosal and submucosal inflammatory infiltrates, submucosal edema and expansion of lymphatics, which **did not differ** among different genotypes treated with DSS (w.t., IL1R^{-/-}, Casp1^{-/-} and CCR2^{-/-}) (this is now presented in supplementary Figure 3D). Specifically, there was no reduction in colon inflammation in any of the knock-out mice compared to w.t. DSS-treated mice.
 - 4) To determine if intestinal permeability differed between genotypes upon DSS-treatment, we further gavaged mice with FITC-Dextran and measured FITC-Dextran concentrations in portal vein serum (as a functional measure of intestinal permeability) in DSS-exposed mice of all of the above genotypes. We found **significantly increased but similar portal vein FITC-Dextran concentrations** (reflecting similar increased intestinal permeability) in all DSS-treated mice, regardless of genotype, compared to non-treated mice. This is further evidence that differences in intestinal inflammation and permeability were not responsible for attenuation of PNAC in these knockout mouse models. (This finding is now represented in supplementary Figure 3E).

We have now added these additional experiments requested by the Reviewer in the Results and Discussion sections of the revised manuscript.

Response to Reviewer #2:

We very much appreciate the Reviewer's positive appraisal of the manuscript and the conclusion that this is a robust model of PNAC, in contrast to the comments of Reviewer 1.

Response to major points:

1. All of the gene expression throughout the paper is interpreted by normalization to a single reference gene (Hprt1). This is not fully appropriate given the variety of models, conditions, differences in cell population (due to inflammatory infiltrate), and physiological states of the tissue and cells examined. MIQE guidelines recommend normalizing the geometric mean of at least three reference genes, whose suitability should be determined empirically for the system under study. Indeed, a recent study demonstrated that Hprt levels are significantly altered in response to DSS treatment (Eissa et al., PLoS One 2016, PMID: PMC4886971). All least three additional reference genes should be measured in the samples presented, and the data reanalyzed (expression level of the reference genes can be measured in the same samples without necessarily re-measuring all of the experimental genes) relative to the geometric mean of the new normalizers. Failing this, the gene expression data cannot be interpreted with any degree of confidence.

Response: We agree with the reviewer that inclusion of additional house-keeping genes can increase the fidelity of experimental gene expression data. This might be especially true, when using Sybr Green based assays. However, we want to emphasize that all our gene expression data were obtained using commercially available TaqMan gene

expression assays, which are optimized by the provider (Life Technologies) to be highly specific. Our initial optimization and pilot studies were conducted using TaqMan gene expression assays to identify suitable housekeeping genes and showed that hepatic mRNA expression of Hprt1 did not change in response to DSS treatment or i.p. LPS injection in wild type mice.

In response to the Reviewer's suggestion that multiple house-keeping genes should be examined, we undertook experiments to validate additional house-keeping genes in our model. We performed experiments examining the four suggested house-keeping genes and found that the genes that were suggested in the above referenced paper (which examined highly inflamed colon) were either expressed too low in liver or revealed different amplification characteristics from our target genes (i.e. expression past cycle 35 and amplification slope significantly different from that of target genes). The only house-keeping gene from the above referenced paper that was expressed adequately in the liver and had suitable amplification characteristics was Ee2f. When Ee2f was substituted for Hprt1 as the house-keeping gene, we replicated our previous results and again found significantly reduced mRNA expression of BSEP, FXR, LXR and ABCG8 in wild type DSS/PN mice relative to chow mice, while restoration of expression was again demonstrated in DSS/PN IL1R^{-/-} and CCR2^{-/-} mice relative to wild type DSS/PN mice . (A graphical representation using Ee2f is now included in supplementary Figure 6). These findings replicate the data presented in the manuscript that were obtained with Hprt1 and thus we propose that our data with Hprt1 can be interpreted with a high degree of confidence. We believe that this additional house-keeping gene analysis further demonstrates the reproducibility and rigor of our findings.

We would like to respond to this critique with two additional points:

1). In light of the associated high cost using commercially available TaqMan gene expression assays, we chose to restrict validation of gene expression using additional housekeeper genes to the most important targets referred to in the current manuscript (LXR, FXR, BSEP, ABCG8) and used a set of wild type DSS/PN mice and IL1R^{-/-} DSS/PN mice and CCR2^{-/-} DSS/PN mice (n=3) with Ee2f as an alternative house keeper.

2) It should be pointed out that the authors in the referenced paper have tested gene expression in a highly inflamed colon, which was exposed to high doses (5% DSS) and long term (7day) DSS treatment, which in our own hands causes high levels of degraded RNA extracted from colon; therefore, it is not surprising that the authors of this manuscript found a high degree of RNA instability in the housekeeping genes they analyzed. What is furthermore important to consider is that in these conditions where there is a high degree of housekeeping RNA instability, one might expect a similarly high degree of RNA instability in target genes (which was not analyzed in the referenced paper). We want to stress that there is much less inflammation in livers from DSS and DSS/PN mice than in the colons exposed to high dose DSS referred to in the referenced paper and therefore the likelihood of RNA degradation in DSS/PN livers is low. Indeed, our RNA quality analysis that we routinely perform using Nanodrop technology consistently reveals high quality RNA and, as mentioned above, Ct values do not differ by more than 2-4 cycles. Furthermore, in the referenced paper gene expression was analyzed using SybrGreen, and it is widely accepted that the fidelity of SybrGreen is inferior to the fidelity of TaqMan probe based assays. Moreover, in their graphical representation of a comprehensive housekeeping gene test in the referenced paper, Hprt1, the housekeeping gene used in our studies, was not grouped into the unstable

RNA targets, consistent with our findings in liver and therefore supporting the use of *Hprt1* as a house keeping gene for our studies.

Response to minor points:

- 2. Since the role of FXR in down-regulating CYP7A1 by increasing the expression of SHP (NR0B2) is a well-known and important component of feedback regulation of bile acid synthesis, particularly under conditions where bile acid levels are high in the liver, but strikingly there was no mention of this. It's possible that *Nr0b2* levels are decreased in the PNAC livers due to the decrease in FXR, and that this increases *Cyp7a1*, exacerbating the cholestasis. This would be trivial to test in the existing samples and would be a nice addition.**

Response: We thank the reviewer for this comment which we have now examined. We now include gene expression data of hepatic *SHP* and *Cyp7a* in the revised Manuscript, which show that, indeed, in DSS/PN mice, *SHP* expression is significantly reduced relative to chow mice, and relative to IL1R^{-/-}, Anakinra-treated and CCr2^{-/-} DSS/PN mice. While in wild type DSS/PN mice *Cyp7a* expression is also reduced (by 50%) it is still significantly higher than that found in IL1R^{-/-}, Anakinra-treated and CCr2^{-/-} DSS/PN mice (reduced by 90%). In order to associate a functional relevance of these gene expression data, we now provide in the Manuscript actual bile acid levels that were measured by LC-MS. Indeed, serum levels of the major taurine conjugated bile acids were significantly increased in wild type DSS/PN mice relative to chow mice, and serum bile acid levels were significantly reduced in IL1R^{-/-} and Anakinra treated and CCr2^{-/-} DSS/PN mice. We conclude that reduced expression of *SHP* in DSS/PN mice (due to reduced expression and reduced transcriptional activity of FXR) results in insufficient suppression of *Cyp7a* with subsequent inadequate suppression of bile acid synthesis. Therefore, this failure to reduce bile acid synthesis in DSS/PN mice might contribute to increased bile acid levels and exacerbate the cholestatic injury, especially in the face of reduced expression of BSEP. These findings are now included in Figure 2A,B in the main manuscript. A statement has been added to the Results and Discussion to reflect these new data and conclusions.

Additional minor points:

Fig 1C Data described as relative mRNA, but it's not clear what they are relative to (chow, WT?). It would seem more sensible to normalize the RQs to DSS/PN WT, at least in this panel.

Response: We have corrected this to clarify that we have normalized Relative mRNA to untreated chow of each (i.e. respective) genotype.

Line 312: Fig 4,E F text and axis label refer to ABCC2, but the figure legend refers to NR1H4. Please correct the discrepancy.

Response: We have made this correction.

Line 318: The use of the word "restored" with respect to the DSS/PN *Il1r*^{-/-} mice may not be appropriate, since they are protected from PNAC from the outset. I suggest

changing it to something more neutral like “Fxr binding to the Abcb11 promoter in DSS/PN II1r -/- mice was comparable to that of chow-fed WT controls”.

Response: We have made this correction.

Multiple instances throughout: “Tukey” misspelled as “Tuckey”.

Response: We have made this correction.

Fig 4, Fig 5 schematics of the relevant promoters and response elements should be provided.

Response: We have included an “Exemplary schematic depicting relative distance between binding sites for FXR and NfκB within Abcb11 promoter” in Figure 4I.

Line 457 should read “sterolin” not “sertolin”.

Response: We have made this correction.

Figure 5F should be presented as the qPCR result, not the qualitative gel images. I suggest cutting the gel image to supplemental material from both Fig 4 and Fig 5.

Response: Since our gel data present a nice way of displaying raw data we would like to include these in the main of the manuscript as they nicely allow the reader to have 2 separate ways of looking at the data within the same figure.

Line 611 Supplier, product number, and (if possible) lot number should be included for the ChIP antisera.

Response: We have made the correction.

Line 613 primer sequences must be included as a supplemental table.

Response: All our PCRs are conducted using commercially available gene expression assays from LifeTechnologies and these can be ordered anytime by anyone from lifeTechnologies, their assays are proprietary. Upon review of the literature it seems customary that it is reported as we did report it herein.

Line 637 What constitutes a “reasonable request”?

Response: We have changed this to “upon request”.

General comments: the presence of several convoluted run-on sentences made parts of the text unnecessarily challenging to read. The figures could be improved by increased attention to detail with respect to stylistic consistency: consistent line weights, axis number formats, axis alignment, axis label rotation.

Response: We have attempted to rewrite and simplify many sections of the manuscript and attempted to improve the figures.

Response to reviewer #3:

We appreciate that the Reviewer considers the current data as extending our knowledge about the cellular and molecular mechanisms of PNAC.

Response to reviewer's concerns:

- 1. In figure 2 the authors demonstrate the functional impact of macrophage derived IL1-beta on expression of hepatic bile acid transporters. The presented data argues that IL1-beta is the predominant inflammatory cytokine inducing PNAC. However, the downstream target NFkB is activated by a variety of different inflammatory mediators. Earlier data showed that Tnf-alpha is an important driver of endotoxin-induced cholestasis. It is essential to include functional Tnf-alpha data to demonstrate that IL1-beta is really the only mediator of this effect in PNAC to better understand the present data compared with earlier published data. Further, it is important to delineate the molecular mechanism of inflammatory macrophage activation. The current data show that cleavage of pro-IL1beta by Caspase1/11 is important. Based on the current data, it seems likely that the Nlrp3 inflammasome mediates PNAC. Gene expression analyses and staining of Nlrp3-sensor molecule are required to address this question.**

Response: We have used a combination of multiple genetic and pharmacological approaches that demonstrate a role for IL1b and macrophages in this model:

- a) DSS/PN mice lacking CCR2 were protected from PNAC. As has been reported in multiple publications, these mice lack the ability to recruit macrophages to the liver and other organs in inflammatory conditions (PMID: 24890723; PMID: 19441102; PMID: 23467934). Therefore, recruited macrophages appear to play a role in this model.

Furthermore, it has been shown that mice lacking Caspase1/11 completely lack the ability to generate IL1-beta (PMID: 7535475). Using these mice, we demonstrate that the final and functional level of the inflammasome pathway to produce mature IL1-beta (i.e. cleavage of pro-IL-1b to mature IL1b by Caspase 1) plays a role in this model. While it could certainly be interesting and informative, it was not a primary goal of our studies to evaluate all of the upstream inflammasome components (of which there exist several) nor the upstream inflammasome inducers in this model. We felt that dissecting the inflammasome would be beyond the scope of the current manuscript. In addition, mRNA expression of NLRP3 is not a predictor of IL1-beta synthesis, which is strictly downstream of activated Caspase1. Therefore, we focused on Caspase 1 and demonstrated by Western blot that Caspase-1 is indeed activated and serum IL-1beta is increased in DSS/PN mice. Therefore, we chose to use Casp1^{-/-} mice as an additional strategy to demonstrate that synthesis of IL1 β plays a role in this model, rather than attempting to determine which inflammasome components and inducers play a role in this model.

- b) We appreciate the comment that other cytokines might be involved in the pathogenesis of PNAC in this mouse model, however, the genetic and pharmacological data that we presented point towards an important role of IL1b, irrespective of the potential role of other cytokines. We agree with the reviewer that in sepsis associated cholestasis, TNFa plays an important role. We want to point out that this PNAC model is however not a sepsis associated cholestasis model. This is demonstrated by the observation that DSS treatment alone (which increases portal

vein LPS levels) does not result in cholestasis in our model, indicating that these LPS levels are likely lower than LPS levels observed in sepsis. Moreover, while DSS treatment resulted in a minor degree of suppression of BSEP in the liver, this degree of BSEP suppression was insufficient to result in increased serum bile acid levels. Specifically, our findings have revealed that inflammatory pathways are most strongly activated when soy-lipid containing PN is administered followed DSS induced intestinal injury. We therefore propose that LPS mediated increases in IL1b promoted some degree of BSEP suppression (which however was insufficient to result in cholestasis) and that IL1b mediated ABCG5/8 suppression allowed PN-derived phytosterols to antagonize FXR, resulting in additional BSEP suppression which was then sufficient to result in cholestasis. Therefore, the pathogenesis of PNAC differs from that of LPS mediated cholestasis. We have included a reference to this in the revised Discussion.

- c) Because it has previously been suggested that IL1b can promote TNF expression (PMID: 1320950; PMID: 21478880; PMID: 2113076) and shown that blocking IL1 receptor reduced TNF (PMID: 8048540), we performed an additional experiment in which we tested expression of *TNFA* in liver homogenate and in isolated MNCs from WT or IL1R^{-/-} mice 4 hours after LPS injection. Consistent with above cited references, we found that in IL1R^{-/-} mice LPS was less effective in inducing TNFA expression in both whole liver and purified MNCs, consistent with the notion that IL1b signaling can function upstream of TNFA in a feedforward mechanism. These new data (depicted in Supplementary Figure 1J,K) are consistent with IL1b playing a dominant upstream role in the regulation of bile transporters in mice with PNAC. Furthermore, a recent study has similarly isolated IL1b and not TNF in mediating alcohol induced liver injury (PMID: 22945633).

2. **Previous reports by the group have already pointed to the role of hepatic macrophages in PNAC. In the first part of their manuscript they further aim to define those hepatic macrophages. To this end, mRNA expression analyses of *Ccl2*, *Ccr2*, *Itgam* and *Emr1* were performed and found to be significantly increased compared to controls. Indeed, the *Ccr2-Ccl2* axis plays an essential role in macrophage recruitment upon liver damage. However, gene expression analyses by far do not suffice to thoroughly phenotype the hepatic immunological response upon PNAC injury. To address this issue, I would highly recommend multicolor flow cytometry analyses of whole liver homogenate using a staining panel to differentiate granulocytes, Kupffer cells, different subsets of monocyte derived macrophages (MoMF) and monocytes. Additional analyses of macrophage activation state would be interesting.**

Response: We agree with the reviewer that a detailed analysis of the hepatic macrophage phenotype in this model would be very informative and interesting and that a more detailed FACS analysis would strengthen our conclusions. We have, therefore, as suggested by the reviewer, performed FACS analysis on whole liver homogenate from mice subjected to various treatments of our model. Using a macrophage staining protocol that we have recently published in *Journal of Immunology* (PMID: 28500078), we now depict the percentages of CD11b high (which represent newly recruited macrophages) vs. F4/80/CD11b low (mainly presenting resident hepatic macrophages (Kupffer cells) macrophage populations in the liver of chow, DSS/chow and DSS/PN mice. These analyses show an increase in the F4/80/CD11b low population in DSS/PN mice relative to chow and DSS/chow mice (now depicted Supplementary Figure 1B) These new data suggest that DSS with subsequent PN infusion results in an expansion

of F4/80/CD11b liver macrophages. It remains to be determined in future more sophisticated tracer studies if this expansion of F4/80/CD11b liver macrophages results from the recruitment of CCR2 positive circulating macrophages. Nevertheless, our data using CCR2^{-/-} mice suggest a role for CCR2 and therefore recruitment of macrophages to the liver in this model.

- 3. Figure 4C shows that reduced Nr1h4 mRNA expression after i.p. injection of LPS in WT mice is attenuated in mice that had undergone treatment with clodronate liposomes. In contrast to the author's conclusion, this treatment does not deplete all hepatic macrophage subsets (and is not specific for hepatic macrophages as basically all phagocytic cells are affected). It is essential to delineate the contribution of individual myeloid cell subsets (Kupffer cells vs. MoMFs vs. monocytes) using flow cytometry to understand the cellular mechanism of the PNAC associated inflammatory response. Gadolinium chloride could be used to specifically target hepatic Kupffer cells.**

Response: It should be pointed out that the clodronate experiments were performed to demonstrate that hepatic mononuclear cells are required to mediate the effect of LPS on suppression of hepatic transporters and that these experiments were not performed in the PNAC mouse model. However, we did provide F4/80 PCR data and F4/80 histochemical analysis of liver that confirm the paucity of macrophages in the liver after clodronate treatment. Furthermore, clodronate has been the method of choice to deplete tissue macrophages in the liver in several recent reports (PMID: 22871793; PMID: 27022031). Nevertheless, as outlined in 2. above, we have now characterized the macrophage population in liver homogenate by flow cytometry in the PNAC mouse model, as suggested by the reviewer. We do not believe that additional experiments are necessary to elucidate the hepatic macrophage populations in response to i.p. LPS administration alone, as this is beyond the scope and questions addressed in this manuscript.

- 4. Reduced inflammatory macrophage infiltration in CCR2^{-/-} mice should not be confirmed based on mRNA expression. Flow cytometry analyses or tissue staining against CD45 (as pan leukocyte marker), CD11b, F4/80 and Ly6G are needed.**

Response: We agree that more robust confirmation of reduced macrophages in CCR2^{-/-} mice would strengthen the Manuscript. We now provide a F4/80 immunohistochemical staining of liver that demonstrates reduced macrophage accumulation in CCR2^{-/-} DSS/PN14d mice (Supplementary Figure 1C,D) and now refer to this in the Results section. In addition, it has been demonstrated by others that CCR2^{-/-} mice have reduced recruitment of inflammatory macrophages (PMID: 28940700).

- 5. The author's conclusion - in line 189-191 - that pathways associated with macrophage activation suppressed expression of bile acid transporters can not be reached based on the presented data. Indeed, it would strengthen the manuscript to analyze the activation status of hepatic macrophages and characterize the (most likely) macrophage driven inflammatory response. However, the current data neither answers which macrophage subset is involved**

(Kupffer cells, different MoMF subsets, classical monocytes (Ly6Chi, CCR2+, CX3CR1low), Ly6Clow patrolling monocytes) nor does it further characterize the activation state of these cells.

Response: Our manuscript provides several lines of evidence that macrophages are involved in suppression of bile and sterol transporters. First, using CCR2^{-/-} mice we demonstrated protection from PNAC and restoration of bile and sterol transporter expression. It has been well established that CCR2^{-/-} mice lack recruitment of macrophages. These mice have been used widely to demonstrate an important role for macrophage pathways in a variety of inflammatory models. Moreover, macrophages have been recognized as the principal generators of IL1 beta, which we have shown in IL1R^{-/-} mice to be the major cytokine involved in PNAC in this study. However, to lend further support to this hypothesis we now show a new Supplementary Figure 1B depicting CD11b high and F4/80/CD11b low macrophages has now been added to the revised manuscript.

6. In a translational perspective, it would be valuable to confirm the proposed mechanism in human PNAC samples (Serum IL1-beta and endotoxin, tissue staining, gene expression analyses).

Response: We agree with the reviewer that evidence of similar pathogenic mechanisms in human PNAC samples would be valuable. These data have been published recently by investigators from Finland who perform protocol liver biopsies in children with PNAC and intestinal failure. These investigators have recently reported gene expression, biochemical analyses and histology of these liver biopsies from children with PNAC (Ann Surg. 2017 Feb 23 PMID: 28234635 and JPEN J Parenter Enteral Nutr. 2016 Mar 9 PMID: 26962062). Importantly, this group's most recent publication demonstrates very similar findings in humans with PNAC to those in our mouse model, with the demonstration of decreased hepatic expression compared to controls of ABCG5/8 and BSEP, correlating with the presence of increased liver histological inflammation, increased inflammatory cytokines including IL1, TNF and IL6, as well as increased serum and liver phytosterol levels. These new data from humans with IFALD/PNAC support the translational significance of the findings in our mouse model. We have added a paragraph to the Discussion these translational findings in humans generated by other investigators.

Reviewers' comments:

Reviewer #1 (Remarks to the Author):

I have no further comments.

Reviewer #2 (Remarks to the Author):

None

Reviewer #3 (Remarks to the Author):

In their manuscript, El Kasmi et al. investigate the role of macrophage mediated IL1-beta/NFkB signaling during parenteral nutrition associated cholestasis (PNAC). The current manuscript builds up on their published in-vivo model of PNAC. These earlier data gave compelling mechanistic evidence that phytosterol promotes liver injury in PNAC. Their earlier work further demonstrated that gut-derived microbiota associated molecular patterns (MAMPs) and TLR4 activation due to intestinal injury represent essential prerequisites for PNAC. However, the model has been published some time ago and is not novel.

In their current manuscript, they aim to further elucidate the cellular and molecular mechanisms involved in PNAC. In the revised version, the authors addressed some of the issues and open questions. However, there are still issues that are not performed in depth.

1. Importantly, the authors now included flow cytometry analyses of whole liver homogenate. Although these data are very important and strengthen the manuscript, it is essential to improve the analyses and presentation of these data in the manuscript. The revised Figure 1 simply shows a bar chart depicting relative abundances of different macrophage subsets. At present, the gating strategy is not clearly described and thus not clear which populations are really analyzed. Further, it is also important to show how neutrophil granulocytes are affected by gating on CD11b+ Ly6G+ cells. Absolute cell quantification would be preferable.

2. Reduced inflammatory macrophage infiltration in CCR2^{-/-} mice should be analyzed by FACS analyses. In their point-by-point response, the authors state that it has already been shown that CCR2^{-/-} mice have reduced recruitment of inflammatory macrophages. However, in the revised Supplemental Figure 1 they now show that these cells are actually not involved in PNAC. Instead, they observe an increase in the CD11b⁺F4/80⁺ population. The authors have to provide a clear gating strategy and include CCR2^{-/-} data to allow the reader to interpret the data and make valid conclusions. Further, the shown CD11b⁺F4/80⁺ population is actually not significantly increased upon PN/DSS. Based on the current data it remains elusive which macrophage subset is involved in the phenotype and how CCR2 deficiency affects different populations in this model.

3. In Point 2 reviewer 1 asked about the novelty of the findings presented in the current manuscript. I would like to point out that the presented mouse model of PNAC itself has already been published by the authors over 5 years ago (Hepatology 2012, Sci Transl Med. 2013). These publications already implicated hepatic macrophages in disease pathogenesis. Hence, the model and postulated mechanism itself are of questionable novelty.

Minor point:

1. Figure 2 F: The western blot lacks a proper loading control. An unspecific band might indicate equal loading, but for publication of reproducible data, a proper loading control is obligatory.

Reviewer #3 (Remarks to the Author):

“In their manuscript, El Kasmi et al. investigate the role of macrophage mediated IL1-beta/NFkB signaling during parenteral nutrition associated cholestasis (PNAC). The current manuscript builds up on their published in-vivo model of PNAC. These earlier data gave compelling mechanistic evidence that phytosterol promotes liver injury in PNAC. Their earlier work further demonstrated that gut-derived microbiota associated molecular patterns (MAMPs) and TLR4 activation due to intestinal injury represent essential prerequisites for PNAC. However, the model has been published some time ago and is not novel.

In their current manuscript, they aim to further elucidate the cellular and molecular mechanisms involved in PNAC. In the revised version, the authors addressed some of the issues and open questions. However, there are still issues that are not performed in depth.”

1.” *Importantly, the authors now included flow cytometry analyses of whole liver homogenate. Although these data are very important and strengthen the manuscript, it is essential to improve the analyses and presentation of these data in the manuscript. The revised Figure 1 simply shows a bar chart depicting relative abundances of different macrophage subsets. At present, the gating strategy is not clearly described and thus*

not clear which populations are really analyzed. Further, it is also important to show how neutrophil granulocytes are affected by gating on CD11b+ Ly6G+ cells. Absolute cell quantification would be preferable.”

Response: We thank the reviewer for their insights and would like to re-address our gating strategy to further clarify our findings. We used a “dump-positive” gate comprised of a mixture of antibodies (CD3, B220, Ly6G) to identify T-cells, B-cells, and Neutrophils that were **excluded** from further analysis. We then took our “dump-negative” gate and stained for CD45, CD11b, and F4/80. Kupffer cells, which have been shown to stain positively, but lowly for CD11b were also identified by their strong F4/80 staining (Dump^{Negative}, CD45⁺, F4/80⁺, CD11b^{Lo}). Recruited macrophages instead stained positively for F4/80 and for CD11b (Dump^{Negative}, CD45⁺, F4/80⁺, CD11b^{Hi}). Finally, since neutrophils were omitted from downstream analysis we identified monocytes by their very weak staining for F4/80 but strong staining for CD11b (Dump^{Negative}, CD45⁺, F4/80⁻, CD11b^{Hi}). In our hands we find this a better strategy to mark for monocytes as Ly6C often segregates into several intermediate populations. A more complete description of this gating strategy has been added to the Methods section on pages 35-36 and in the Legend for Supplemental Figure 1.

2. “Reduced inflammatory macrophage infiltration in CCR2-/- mice should be analyzed by FACS analyses. In their point-by-point response, the authors state that it has already been shown that CCR2-/- mice have reduced recruitment of inflammatory macrophages. However, in the revised Supplemental Figure 1 they now show that these cells are actually not involved in PNAC. Instead, they observe an increase in the CD11b+F4/80+ population. The authors have to provide a clear gating strategy and include CCR2-/- data to allow the reader to interpret the data and make valid conclusions. Further, the shown CD11b+F4/80+ population is actually not significantly increased upon PN/DSS. Based on the current data it remains elusive which macrophage subset is involved in the phenotype and how CCR2 deficiency affects different populations in this model.”

Response: We believe our findings concerning the recruitment of macrophages and resident tissue macrophages is backed by findings described in other organ systems under settings of inflammation (e.g. PMID: 25319326). In Supplemental Figure 1B, we have now amended the axes to represent the results more clearly, we have added in the percentage of Kupffer cells, recruited macrophages and monocytes, and we have created a figure that summarizes our gating strategy (in Suppl Figure 1B). We would like to highlight that it appears clear that KCs are “contracting” at the expense of an influx of recruited macrophages measured as a percentage of CD45+ cells, and therefore we do not believe that an absolute quantification, as suggested by the reviewer, would add further insight. Further evaluation to provide definitive answers to the complex question of lineage fate for monocytes/macrophages in liver injury models requires labor intense lineage tracing studies, which we consider beyond the scope of testing the hypothesis of this manuscript. Our findings are consistent with a recent study by Zigmond et al (PMID 24890723), which show a contraction of Kupffer cells at the expense of recruitment of macrophages in a CCR2 dependent pathway and these cells appear to convert very quickly into F4/80 positive cells. Moreover, it is known that CCR2 deficient mice lack emigration of Ly6C high cells from the bone marrow and therefore using CCR2 deficient mice for flow cytometry testing would not add any further information regarding CCR2 and Ly6C in this model. We believe the best strategy to define the role of CCR2 positive

macrophages is to test if PNAC is attenuated in CCR2 deficient mice, which is reported here in this manuscript. This observation of protection from PNAC in CCR2^{-/-} DSS/PN mice argues strongly for an involvement of CCR2 positive cells in this model. It is widely accepted that CCR2 is expressed on recruited CD11b positive (PMID: 27990288) macrophages and thus this is congruent with the FACS data and functional data in the CCR2^{-/-} mice provided in this manuscript. Additionally, we believe panel 1C and 1D demonstrates the effect aforementioned since recruited macrophages in our FACS and IHC data express F4/80 and these panels serve as a visual representation for the influx of recruited macrophages that is dissipated in CCR2^{-/-} mice.

3. *“In Point 2, reviewer 1 asked about the novelty of the findings presented in the current manuscript. I would like to point out that the presented mouse model of PNAC itself has already been published by the authors over 5 years ago (Hepatology 2012, Sci Transl Med. 2013). These publications already implicated hepatic macrophages in disease pathogenesis. Hence, the model and postulated mechanism itself are of questionable novelty.”*

Response: We thank the reviewer for citing our earlier publications on TLR4 and phytosterols in this model and agree that the model was published 5 years ago. In the current manuscript, the novelty lies in the fact that the molecular and cellular pathways in this PNAC model are mechanistically delineated and open up possibilities for treatment using currently FDA approved compounds, such as anakinra, or targeting other pro-inflammatory pathways downstream of TLR4 and IL1 signaling, such as convergence points downstream of TLR4 and IL1, such as MyD88, Irak4, Irak1, etc. Moreover, the fact that our study highlights the molecular interplay of FXR and LXR with inflammatory NFκB signaling and how this interplay regulates both bile and sterol transport on a molecular level is novel in considering the pathogenesis of PNAC. Our study is also novel in that it has revealed potential molecular pathways in the cellular cross talk between macrophages and hepatocytes in the pathogenesis of PNAC, which has not been addressed at all in any previous study. Despite these findings, we are happy to remove the word “novel” before the words “mouse model” as suggested by the reviewer.

Minor point:

1. *“Figure 2 F: The western blot lacks a proper loading control. An unspecific band might indicate equal loading, but for publication of reproducible data, a proper loading control is obligatory.”*

Response: We respectfully disagree with this point because the non-specific binding of the Ab used does show equal loading similar to what an Ab raised against a house keeping gene would show. This is documented by previous publications by other authors (e.g., PMID 14568929 Figure 1 C). However, we have now added an additional loading control, as suggested by the reviewer, using anti-GRB2 antibody which shows similar loading of all lanes.

Reviewer #3 (Remarks to the Author):

“In their manuscript, El Kasmi et al. investigate the role of macrophage mediated IL1-beta/NFkB signaling during parenteral nutrition associated cholestasis (PNAC). The current manuscript builds up on their published in-vivo model of PNAC. These earlier data gave compelling mechanistic evidence that phytosterol promotes liver injury in PNAC. Their earlier work further demonstrated that gut-derived microbiota associated molecular patterns (MAMPs) and TLR4 activation due to intestinal injury represent essential prerequisites for PNAC. However, the model has been published some time ago and is not novel.

In their current manuscript, they aim to further elucidate the cellular and molecular mechanisms involved in PNAC. In the revised version, the authors addressed some of the issues and open questions. However, there are still issues that are not performed in depth.”

1.” Importantly, the authors now included flow cytometry analyses of whole liver homogenate. Although these data are very important and strengthen the manuscript, it is essential to improve the analyses and presentation of these data in the manuscript. The revised Figure 1 simply shows a bar chart depicting relative abundances of different macrophage subsets. At present, the gating strategy is not clearly described and thus not clear which populations are really analyzed. Further, it is also important to show how neutrophil granulocytes are affected by gating on CD11b+ Ly6G+ cells. Absolute cell quantification would be preferable.”

Response: We thank the reviewer for their insights and would like to re-address our gating strategy to further clarify our findings. We used a “dump-positive” gate comprised of a mixture of antibodies (CD3, B220, Ly6G) to identify T-cells, B-cells, and Neutrophils that were excluded from further analysis. We then took our “dump-negative“ gate and stained for CD45, CD11b, and F4/80. Kupffer cells, which have been shown to stain

positively, but lowly for CD11b were also identified by their strong F4/80 staining (DumpNegative, CD45+, F4/80+, CD11bLo). Recruited macrophages instead stained positively for F4/80 and for CD11b (DumpNegative, CD45+, F4/80+, CD11bHi). Finally, since neutrophils were omitted from downstream analysis we identified monocytes by their very weak staining for F4/80 but strong staining for CD11b (DumpNegative, CD45+, F4/80-, CD11bHi). In our hands we find this a better strategy to mark for monocytes as Ly6C often segregates into several intermediate populations.

2. "Reduced inflammatory macrophage infiltration in CCR2^{-/-} mice should be analyzed by FACS analyses. In their point-by-point response, the authors state that it has already been shown that CCR2^{-/-} mice have reduced recruitment of inflammatory macrophages. However, in the revised Supplemental Figure 1 they now show that these cells are actually not involved in PNAC. Instead, they observe an increase in the CD11b⁺F4/80⁺ population. The authors have to provide a clear gating strategy and include CCR2^{-/-} data to allow the reader to interpret the data and make valid conclusions. Further, the shown CD11b⁺F4/80⁺ population is actually not significantly increased upon PN/DSS. Based on the current data it remains elusive which macrophage subset is involved in the phenotype and how CCR2 deficiency affects different populations in this model."

Response: We believe our findings concerning the recruitment of macrophages and resident tissue macrophages is backed by findings analogous in other organ systems under settings of inflammation (e.g. PMID: 25319326). In Supplemental Figure 1B, we have now amended the axis to represent the results more clearly, we have added in the percentage of Kupffer cells, and we have created a figure that summarizes our gating strategy (Suppl Figure 1B). We would like to highlight that it appears that KCs are "contracting" at the expense of an influx of recruited macrophages measured as a percentage of CD45⁺ cells, and therefore we do not believe that an absolute quantification, as suggested by the reviewer, would add further insight. Definitive answers to this complex question of lineage fate in liver injury models requires labor intense lineage tracing studies which we consider beyond the scope of testing the hypothesis of this manuscript. Our findings are consistent with a recent study by Zigmond et al (PMID 24890723), which show a contraction of Kupffer cells at the expense of recruitment of macrophages in a CCR2 dependent pathway and these cells appear to convert very quickly into F4/80 positive cells. Moreover, it is known that CCR2 deficient mice lack emigration of Ly6C high cells from the bone marrow and therefore using CCR2 deficient mice does not add any further information regarding CCR2 and Ly6C in this model. The best experiment to do is to test if PNAC is attenuated in CCR2 deficient mice, exactly as reported here in this manuscript. This observation of protection from PNAC in CCR2^{-/-} DSS/PN mice argues strongly for an involvement of CCR2 positive cells in this model. It is widely accepted that CCR2 is expressed on recruited CD11b positive (PMID: 27990288) macrophages and thus this is congruent with the FACS data and functional data in the CCR2^{-/-} mice provided in this manuscript. Additionally, we believe panel 1C and 1D demonstrates the effect aforementioned since recruited macrophages in our FACS and IHC data express F4/80 and these panels serve as a visual representation for the influx of recruited macrophages.

3. "In Point 2 reviewer 1 asked about the novelty of the findings presented in the current manuscript. I would like to point out that the presented mouse model of PNAC itself has already been published by the authors over 5 years ago (Hepatology 2012, Sci Transl Med. 2013). These publications already implicated hepatic macrophages in disease

pathogenesis. Hence, the model and postulated mechanism itself are of questionable novelty.”

Response: We thank the reviewer for citing our earlier publications on TLR4 and phytosterols in this model. In the current manuscript, the novelty lies in the fact that the molecular and cellular pathways in this model have been mechanistically delineated and open up possibilities for treatment using currently FDA approved compounds, such as Anakinra, or targeting other pro-inflammatory pathways downstream of TLR4 and IL1 signaling, such as convergence points downstream of TLR4 and IL1, such as MyD88, Irak4, Irak1, etc. Moreover, the fact that our study has highlighted the molecular interplay of FXR and LXR with inflammatory NFkB signaling and how this interplay regulates both bile and sterol transport on a molecular level is highly novel in considering the pathogenesis of PNAC. Our study is also novel in that it has revealed potential molecular pathways in the cellular cross talk between macrophages and hepatocytes in the pathogenesis of PNAC, which has not been addressed at all in any previous study. Despite these findings, we are happy to remove the word “novel” before the words “mouse model” as suggested by the reviewer.

Minor point:

1. “Figure 2 F: The western blot lacks a proper loading control. An unspecific band might indicate equal loading, but for publication of reproducible data, a proper loading control is obligatory. “

Response: We respectfully disagree with this point because the non-specific binding of the Ab used does show equal loading just as much as any other Ab raised against a house keeper would show. This is documented by previous publications by other authors (e.g., PMID 14568929 Figure 1 C). However, we have now added an additional loading control using anti-GRB2 antibody.